# Underestimation of Column NO₂ Amounts from the OMI Satellite Compared to Diurnally Varying Ground-Based Retrievals from Multiple Pandora Spectrometer Instruments

Jay Herman[1], Nader Abuhassan[1], Jhoon Kim[2], Jae Kim[3], Manvendra Dubey[4], Marcelo Raponi[5], Maria Tzortziou[6]

## Abstract

Retrievals of Total Column NO₂ ($TCNO_2$) are compared for 14 sites from the Ozone Measuring Instrument (OMI using OMNO2-NASA v3.1) on the AURA satellite and from multiple ground-based PANDORA spectrometer instruments making direct-sun measurements. While OMI accurately provides the daily global distribution of retrieved $TCNO_2$, OMI almost always underestimates the local amount $TCNO_2$ by 50 to 100% in polluted areas, while occasionally the daily OMI value exceeds that measured by PANDORA at very clean sites. Compared to local ground-based or aircraft measurements, OMI cannot resolve spatially variable $TCNO_2$ pollution within a city or urban areas, which makes it less suitable for air quality assessments related to human health. In addition to systematic underestimates in polluted areas, OMI's selected 13:30 equator crossing time polar orbit causes it to miss the frequently much higher values of $TCNO_2$ that occur before or after the OMI overpass time. Six discussed Northern Hemisphere PANDORA sites have multi-year data records (Busan, Seoul, Washington DC, Waterflow New Mexico, Boulder Colorado, and Mauna Loa) and one site in the Southern Hemisphere (Buenos Aires Argentina). The first four of these sites and Buenos Aires frequently have high $TCNO_2$ ($TCNO_2 >$ 0.5 DU). Eight additional sites have shorter term data records in the US and South Korea. One of these is a one-year data record from a highly polluted site at City College in New York City with pollution levels comparable to Seoul, South Korea. OMI estimated air mass factor, surface reflectivity, and the OMI 24x13 km$^2$ FOV (field of view) are three factors that can cause OMI to underestimate $TCNO_2$. Because of the local inhomogeneity of $NO_x$ emissions, the large OMI FOV is the most likely factor for consistent underestimates when comparing OMI $TCNO_2$ to retrievals from the small PANDORA effective FOV (measured in m$^2$) calculated from the solar diameter of 0.5$^O$.

Key Words: Nitrogen dioxide, OMI, PAN, PANDORA, ground-based, satellite

Correspondence email: jay.r.herman@nasa.gov

[1]University of Maryland Baltimore County JCET, Maryland
[2]Department of Atmospheric Sciences, Yonsei National University, South Korea
[3]Department of Atmospheric Science, Pusan National University, South Korea
[4]Earth Systems Observations, Los Alamos National Laboratory, Los Alamos, NM 87545
[5]Departamento de Investigaciones en Láseres y Aplicaciones (DEILAP), Instituto de Investigaciones Científicas y Técnicas para la Defensa (CITEDEF), Ministerio de Defensa (MINDEF), Buenos Aires, Argentina
[6]City College of New York, New York City, NY

**Underestimation of Column NO₂ Amounts from the OMI Satellite Compared to Ground-Based Retrievals from Multiple Pandora Spectrometer Instruments**

## 1.0 Introduction

Retrieval of Total Column $NO_2$ ($TCNO_2$) from the Ozone Monitoring Instrument (OMI) has been a scientific success story for the past 14 years. Near total global coverage from the well-calibrated OMI has enabled observation of all the regions where $NO_2$ is produced and has permitted monitoring of the changes during the 2004 to 2019 period, especially in regions where there is heavy and growing industrial activity (e.g., China and India). $TCNO_2$ amounts (data used: OMNO2-NASA v3.1) retrieved from OMI over various specified land locations show a strong local underestimate compared to co-located PANDORA Spectrometer Instruments (the abbreviation PAN is used for graph and table labels). The underestimate of OMI $TCNO_2$ at the overpass time compared to ground-based measurements has previously been reported at a few specific locations (Bechle, 2013; Lamsal et al., 2015; Ialongo et al., 2017; Kollonige, et al., 2018; Goldberg et al., 2018; Herman et al., 2018). The accuracy and precision of PANDORA $TCNO_2$ measurements has been previously discussed (Herman et al., 2009; 2018). For any location, the OMI overpass local standard time consists of the central overpass near the 13:30 hour equator crossing solar time and occasionally a side viewing overpass from adjacent orbits within ±90 minutes of the central overpass time. Independently from instrument calibration and retrieval errors, there are two specific aspects to the underestimation of $TCNO_2$ pollution levels. Because of OMI's selected polar orbit, it is not possible for the mid-day OMI observations to see the large diurnal variation of $TCNO_2$ that usually occur after the 13:30 overpass time, and second, because of spatial inhomogeneity the large OMI field of view (FOV) footprint 13 x 24 $km^2$ at OMI nadir view tends to average regions of high $NO_2$ amounts (Nowlan et al., 2016; Judd et al., 2018) with those from lower pollution areas. An analysis by Judd et al., (2019, their Fig. 9) shows the effect of decreasing satellite spatial resolution on improving agreement with PANDORA, with the best agreement occurring with an airborne instrument, GEO-TASO (resolution 3x3 $km^2$) followed by TropOMI (5x5 $km^2$) and then OMI (18x18 $km^2$). Both OMI and TropOMI show an underestimate of $TCNO_2$ compared to PANDORA.

There are other possible systematic retrieval errors with OMI $TCNO_2$. The largest of these is determining the air mass factor (AMF) needed to convert slant column measurements into vertical column amounts followed by the surface reflectivity Rs (Boersma et al., 2011; Lin et al., 2015; Nowlan et al., 2016; Lorente et al., 2018). Accurately determining the AMF for $TCNO_2$ requires a-priori knowledge of the $NO_2$ profile shape (Krotkov et al., 2017), which is estimated from coarse resolution model calculations (Boersma et al., 2011), and using the correct Rs. Currently Rs is found using a statistical process of sorting through years of data to find relatively clear-sky scenes for each location (Kleipool, et al., 2008; O'Byrne et al., 2010). Boersma et al., 2004 gave a detailed error analysis for the various components contributing OMI $TCNO_2$ retrievals resulting an estimated "retrieval precision of 35-60%" in heavily polluted areas dominated by determining the air mass factor. An improved V2.0 DOMINO retrieval (Boersma et al., 2011) algorithm reduced the retrieval errors while increasing the estimated airmass factor, which reduces the retrieved $TCNO_2$ up to 20% in winter and 10% in summer. The current version of OMNO2-NASA (Krotkov et al., 2017) and v2.0 DOMINO (Boersma et al., 2011) are generally in good agreement (Marchenko et al.,

2015; Zara et al., 2018). However, the OMNO2-NASA TCNO$_2$ retrievals are 10 to 15% lower than the v2.0
DOMINO retrievals and with Quality Assurance for Essential Climate Variables (QA4ECV) retrievals. A
subsequent detailed analysis of surface reflectivity (Vasilkov et al., 2017) shows that retrieval of TCNO$_2$ in
highly polluted areas (e.g., some areas in China) can increase by 50% with the use of geometry-dependent
reflectivities, but only increase about 5% in less polluted areas. For PANDORA, calculation of the solar
viewing AMF is a simple geometric problem (AMF is approximately proportional to the cosecant of the
solar zenith angle SZA) and is independent of $R_S$ (Herman et al., 2009). For a highly polluted region with
TCNO$_2$ = 5.34x10$^{16}$ molecules/cm$^2$ or 2 DU, the PANDORA error is expected to be less than 2 ± 0.05 DU
(±2.5%) with the largest uncertainty coming from an assumed nominal amount of stratospheric TCNO$_2$ =
0.1 DU.
Accurate satellite TCNO$_2$ retrievals (and for other trace gases) are important in the estimate of
the effect of polluted air containing NO$_2$ on human health (Kim and Song, 2017 and references therein),
especially from the viewpoint of NO$_2$ as a respiratory irritant and precursor to cancer (Choudhari et al.,
2013). Since NO$_2$ is largely produced by combustion, satellite observations of NO$_2$ serve as a proxy for
changing industrial activity. Another important application requiring accurate measurements of the
amount of TCNO$_2$ and its diurnal variation is atmospheric NO$_2$ contribution to nitrification of coastal
waters (Tzortziou et al., 2018).
We show that the use of OMI TCNO$_2$ for estimating local air quality and coastal nitrification on a
global basis is misleading for most polluted locations, and especially on days when the morning or
afternoon amounts are higher than those occurring at the OMI overpass time near 13:30 hours standard
time. OMI TCNO$_2$ data are extremely useful for estimating regional pollution amounts and for assessing
long-term changes in these amounts. Modelling studies (Lamsal et al., 2017 Fig. 1) based on the Global
Modelling Initiative model (Strahan et al., 2007) simulating TCNO$_2$ diurnal variation over Maryland USA
(37-40$^O$N, 74-79$^O$W) shows a late afternoon peak and shows that the stratospheric component does not
substantially contribute to this peak. Boersma et al. (2016) show that sampling strategy can cause
systematic errors between OMI TCNO$_2$ and model TCNO$_2$ with satellite results being up to 20% lower than
models. Duncan et al., (2014) reviews the applicability of satellite TCNO$_2$ data to represent air quality and
notes that TCNO$_2$ correlates well with surface levels of NO$_2$ in industrial regions and states that the portion
of TCNO$_2$ in the boundary layer could be over 75% of the total vertical column depending on NO$_2$ altitude
profile shape.
This paper presents 14 different site comparisons between retrieved OMI TCNO$_2$ overpass values
that are co-located with PANDORA TCNO$_2$ amounts from various locations in the world. Six of the
comparisons are where PANDORAs have long-term data (1-year or longer) records. The comparisons are
done using 80 second cadence data matched to the OMI overpass times averaged over ±6 minutes and
with monthly running averages calculated using Lowess(f) (Locally Weighted least squares fit to a fraction
f of the data points, (Cleveland, 1981) of OMI-PANDORA time matched TCNO$_2$. OMI overpass data,
https://avdc.gsfc.nasa.gov/index.php?site=666843934&id=13, are filtered for the row anomaly and
cloudy pixels. The selection of a $\pm$6-minute window represents 720 seconds or 9 PANDORA
measurements averaged together around the OMI overpass time to reduce the effect of outlier points.
The specific value of ±6 minutes is arbitrary but increases the already high effective signal to noise ratio
by a factor of 3. PANDORA data are filtered for significant cloud cover by examining the effective variance
in sub-interval (20 seconds) measurements. Each PANDORA listed measurement is the average of up to
4000 (clear sky) individual measurement made over 20 seconds.

113       This paper gives a discussion and presentation of data on the effect of diurnal variation that are
always missed at the local OMI mid-day overpass times. We show that OMI $TCNO_2$ values are also
systematically lower than PANDORA values at sites with significant pollution ($TCNO_2 > 0.3$ DU). We present
a unique view of a year of fully time resolved diurnal variation of $TCNO_2$ at two sites, Washington DC and
New York City, which are similar to other polluted locations.

**2.0 Brief Instrument Descriptions**

120       For the purposes of $TCNO_2$ retrievals, both OMI and PANDORA are spectrometer-based
instruments using nearly the same spectral range and similar spectral resolution (about 0.5 nm). Both use
spectral fitting retrieval algorithms that differ (Boersma et al. 2011; Herman et al., 2009) because of the
differences between direct-sun viewing retrievals (PANDORA) and above the atmosphere downward
viewing retrievals (OMI). The biggest difference is with the respective fields of view, 13 x 24 $km^2$ at OMI
nadir view and larger off-nadir FOV compared to the much smaller PANDORA FOV ($1.2^O$) measured in $m^2$
with the precise value depending on the $NO_2$ profile shape and the solar zenith angle. For example, if most
of the $TCNO_2$ is located below 2 km, then the PANDORA FOV is approximately given by
$(1.2\pi/180)(2/\cos(SZA))$, which for SZA = $45^O$ is about 59x59 $m^2$. If the solar disk ($0.5^O$) is used as the limiting
factor, then the effective FOV is smaller (25x25 $m^2$).
**2.1 OMI**

131       OMI is an east-west side (2600 km) and nadir viewing polar orbiting imaging spectrometer that
measures the earth's backscattered and reflected radiation in the range 270 to 500 nm with a spectral
resolution of 0.5 nm. The polar orbiting side viewing capabilities produce a pole to pole swath that is about
2600 km wide displaced in longitude every 90 minutes by the earth's rotation to provide coverage of
nearly the entire sunlit Earth once per day at a 13:30 solar hour equator crossing time with spatial gaps at
low latitudes. OMI provides full global coverage every 2 to 3 days. Additional gaps are caused by a problem
with the OMI CCD, "row anomaly" (Torres et al., 2018) that effectively reduces the number of near-nadir
overpass views. A detailed OMI instrument description is given in Levelt et al. (2006). $TCNO_2$ is determined
in the visible spectral range from 405 to 465 nm where the $NO_2$ absorption spectrum has the maximum
spectral structure and where there is little interference from other trace gas species (there is a weak water
feature in this range). OMI $TCNO_2$ overpass data are available for many ground sites (currently 719) from
the following NASA website. https://avdc.gsfc.nasa.gov/index.php?site=666843934&id=13 (valid as of 16
July 2019).

**2.2 PANDORA**

147       PANDORA is a sun-viewing instrument for SZA < $80^O$ that obtains about 4000 spectra for clear-sky
views of the sun in 20 seconds for each of two ranges UV (290 – 380 nm using a UV340 bandpass filter)
and visible plus UV (280 – 525 nm using no filter). The overall measurement time is about 80 seconds
including a 20 second dark-current measurements between each spectral measurement throughout the
day. About 4000 clear-sky spectra for the UV and visible portions are separately averaged together to
achieve very high signal to noise ratios (SNR). The UV340 filter for UV portion of the spectra reduces stray
light effects from the visible wavelength range.  A detailed description of PANDORA and its SNR is given
in Herman et al., (2009; 2015). The effect of moderate cloud cover (reduction of observed signal by a
factor of 8) in the PANDORA FOV on $TCNO_2$ retrievals is small (Herman et al., 2018). Cloud cover also
reduces the number of measurements possible in 20 seconds, which potentially increases the noise level.
PANDORA is driven by a highly accurate sun tracker that points an optical head at the sun and transmits
the received light to an Avantes 2048 x 32 pixel CCD spectrometer (AvaSpec-ULS2048 from 280 – 525 nm
with 0.6 nm resolution) through a 50 micron diameter fiber optic cable. The estimated $TCNO_2$ error is
approximately 0.05 DU (1 DU = 2.69 x $10^{16}$ molecules cm$^{-2}$) out of a typical value of 0.3 DU in relatively
clean areas and over 3 DU in highly polluted areas. PANDORA data are available for 250 sites. Some sites
have multi-year data sets, but many of these sites are short-term campaign sites.
https://avdc.gsfc.nasa.gov/pub/DSCOVR/Pandora/DATA_01/. (valid as of 16 July 2019).
**3.0 Overpass Comparisons and Diurnal Variation of $TCNO_2$**
The contribution of $NO_2$ to air quality at the Earth's surface is usually a proportional function of
$TCNO_2$ that varies with the time of day and with the altitude profile shape (Lamsal et al., 2013; Bechle et
al., 2013). Most of the $NO_2$ amount is usually located between 0 and 3 km altitude with a small amount of
about 0.1±0.05 DU (Dirksen et al. 2011) in the upper troposphere and stratosphere. Because of the
relatively short chemical lifetime, 3-4 hours (Liu et al., 2016), in the lower atmosphere, most of the $NO_2$ is
located near (0 to 20 km) its sources (industrial activity, power generation, and automobile traffic). At
higher altitudes or in the winter months, the life time of $NO_2$ is longer permitting transport over larger
distances from its sources.
During the South Korean campaign (KORUS-AQ) in the spring of 2016 the diurnal variations of
$TCNO_2$ vs days of the year DOY were determined for 6 sites (Herman et al., 2018), one of which is
reproduced here (Fig. 1) for the city of Busan showing relatively low values of $TCNO_2$ in the morning (0.5
DU), moderately high values during the middle of the day (1.3 DU), and very high values on some of the
afternoons (2 to 3 DU). Of these data, OMI only observes midday values near the 13:30 time marked on
the Local_Time axis of Fig.1 thereby missing very high values (2 to 3 DU) that frequently occur later in the
afternoon coinciding with times when people are outdoors returning from work.
In addition to not being designed to observe the $TCNO_2$ diurnal variation, the OMI values are
about half those observed by PANDORA (Fig. 2) at the OMI overpass time, so that using OMI values to
estimate $NO_2$ pollution seriously underestimates the air quality problem even at midday. The shaded area
in Fig.2 corresponds to the period covered in the KORUS-AQ campaign 7 April to 11 June 2016 shown in
Fig. 1. An extended time series for Busan location is shown in Fig. 3.

Because of the different effective $NO_2$ FOV of PANDORA (measured in $m^2$) while tracking the
moving sun position located in the heart of Busan (FOV distance d < 5 km for an SZA < $70^O$ used for
$TCNO_2$ retrievals), both the daily (Fig. 3, left panel) and PANDORA monthly average variation (Fig. 3, right
panel), obtained at the OMI overpass time, differs from the variation in the OMI $TCNO_2$ caused by the
much larger OMI FOV (13 x 24 $km^2$ at OMI nadir view) retrieval. Because of this, the OMI time series has
low correlation ($r^2$ = 0.1) with the PANDORA time series.
The extended OMI vs PANDORA time series from 2012 – 2017 for Busan (Fig. 3) shows the same
magnitude of differences seen during the KORUS-AQ period. A similar OMI vs PANDORA plot for total
column ozone $TCO_3$ (Appendix Fig A1) shows good agreement between PANDORA and OMI indicating that
the PANDORA instrument was operating and tracking the sun properly. Because the spatial variability of
$TCO_3$, which is mostly in the stratosphere, is much less than for $TCNO_2$, the effect of different FOV's is
minimized for ozone.
The same type of differences, $TCNO_2$(PAN) > $TCNO_2$(OMI), are seen at a wide variety of sites (e.g.,
see Fig.4 and Fig. 5) for Northern Hemisphere sites and one site in the Southern Hemisphere where
PANDORA has an extended time series. Comparing extended Busan multi-year time series, some broad-
scale correlation can be seen with peaks in February 2013, January 2014, and in 2016. The OMI data from
Busan are different than data from many sites, since Busan is located very near the ocean causing a portion
of the OMI FOV to be over the relatively unpolluted ocean areas, whereas PANDORA is located inland
(Pusan University) in an area of dense automobile traffic and quite near mountains capable of trapping
air.
Figures 4 and 5 show a variety of different sites, ranging from the Mauna Loa Observatory location
at 3.4 km (11,161 feet) on a relatively clean Hawaiian Island surrounded by ocean to a polluted landlocked
semi-arid site at Waterflow, New Mexico near a power plant. All the sites considered show a significant
underestimate of OMI $TCNO_2$. A summary of the monthly average underestimates is given in Tables 1 and
2. For some sites there is evident correlation between the two offset measurements. For example, the
PANDORA at NASA Headquarters in Washington DC tracks the OMI measurement quite well on a monthly
average basis with a correlation coefficient of $r^2$(mn) = 0.7 even though the daily correlation is low ($r^2$(dy)
= 0.17). Other sites have only short periods of correlation and overall weak correlation (Table 1 showing
daily, dy and monthly, mn, correlation coefficients for the graphs in Figures 4 and 5)
$TCNO_2$(PAN) comparisons with $TCNO_2$(OMI) from Mauna Loa Observatory MLO (Fig. 4) are not
those that might be expected, since the PANDORA observations are in an area where there are almost no
automobile emissions and certainly no power plants, yet PAN > OMI and $TCNO_2$(PAN) values are large
enough so that the pollution values (0.18 DU) are well above the stratospheric values (approximately 0.1
DU). OMI, which mainly measures values over the clean ocean, has an average value of about 0.1 DU (see
appendix Fig. A2). Since there are no emission or combustion sources of $NO_2$ at high altitudes near MLO
at 3.4 km, the PANDORA values suggest upward airflow from the near sea level circumferential ring road,
Keahole oil power plant, and resort areas. The Mauna Loa $TCNO_2$ values do not show any correlation with
the recent increased volcanic activity at Mt. Kilauea after 2016. A graph showing the mid-day values of
TCNO$_2$ at MLO is given in the appendix. Recently, the original Mauna Loa PANDORA has been replaced.
The new instrument's calibration will be reviewed before being added to the time series as part of a
general data quality assurance program that is starting with the most recently deployed or upgraded
PANDORA instruments at about 100 locations.

Table 1 Values of TCNO$_2$ for PANDORA and OMI from monthly averages in Figs. 4 and 5

| Name | Location (Lat, Lon) | PAN (DU) | OMI (DU) | r$^2$ (dy, mn) |
|---|---|---|---|---|
| Mauna Loa Hawaii | 19.536°, -155.5762° | 0.16 | 0.11 | 0.01, 0.30 |
| NASA HQ Washington DC | 38.882°, -77.01° | 0.34 | 0.25 | 0.17, 0.70 |
| Waterflow New Mexico[1] | 36.797°, -108.48° | 0.32 | 0.18 | 0.13, 0.52 |
| Seoul South Korea | 37.5644°, 126.934° | 1.2 | 0.58 | 0.11, 0.06 |
| Busan South Korea | 35.2353°, 129.0825° | 0.68 | 0.32 | 0.09, 0.10 |
| Boulder Colorado | 39.9909°, -105.2607° | 0.27 | 0.17 | 0.04, 0.09 |
| Buenos Aires Argentina | -34.5554°, -58.5062° | 0.50 | 0.26 | 0.16, 0.08 |
| **Average** | | **0.49** | **0.27** | |


Table 2 Average values of TCNO$_2$ for PANDORA and OMI for additional sites

| Name | Location (Lat, Lon) | PAN (DU) | OMI (DU) |
|---|---|---|---|
| Essex Maryland | 39.31083°, -76.47444° | 0.30 | 0.28 |
| Baltimore Maryland | 39.29149°, -76.59646° | 0.45 | 0.27 |
| Fresno California | 36.7854°, -119.7731° | 0.42 | 0.17 |
| Denver La Casa Colorado | 39.778°, -105.006° | 0.68 | 0.19 |
| GIST[2] | 35.226°,126.843° | 0.42 | 0.20 |
| HUFS[3] | 37.338°,127.265° | 0.61 | 0.51 |
| City College New York City | 40.8153°,-73.9505° | 0.60 | 0.40 |
| **Average** | | **0.50** | **0.29** |

[1]Waterflow, NM is listed for OMI data as Four Corners, NM, a nearby landmark
[2]Gwangju Institute of Science and Technology S. Korea
[3]Hankuk University Foreign Studies South Korea


An interesting inland site is near the very small town of Waterflow, New Mexico (Figs. 4 and 6),
where two power plants located near the PANDORA site ceased operation on December 30, 2013
(Lindenmaier et al., 2014). According to a quote from AZCentral Newspaper (Tuesday 31 December 2013)
"Three coal-fired generators that opened in the 1960s near Farmington, N.M., closed Monday as part of
a $182 million plan for Arizona Public Service Co. to meet environmental regulations, the utility reported".
The TCNO$_2$ data suggests that the actual shutdown occurred near October 15, 2013. After the shutdown,
air quality improved in the area with TCNO$_2$ decreasing from 0.4 DU to 0.28 DU.  The remaining more
efficient generators continued to produce smaller N0$_2$ emissions. These were shut down at the end of
2016 with little additional observed change in TCNO$_2$, since these boilers used NO$_2$ scrubbers (Dubey at
al., 2018 in preparation). A nearby highway (Route 64) about 2 km from the PANDORA site has little
automobile traffic. An example of the diurnal behavior of TCNO$_2$ at Waterflow, New Mexico on 6 June
2012 is shown in Fig. 6 to illustrate the behavior of PANDORA TCNO$_2$ retrievals at a wide range of SZA. The

terrain surrounding the Waterflow Pandora site is flat with no obstructions (buildings) permitting observations to very high SZA. Almost every day the power plant briefly puts out very high emissions of $NO_2$ as part of its daily boiler cleaning cycle. This can be seen in the very high peak value of $TCNO_2$ of 3.4 DU compared to the nominal value of 0.5 DU occurring for most of the day. The value from the FOV averaged OMI retrieval at 21:01 GMT (14:01 local standard time) is about 0.2 DU compared to the PANDORA value of about 0.5DU. Figure 6 also illustrates $TCNO_2$ diurnal behavior at two other sites, NASA HQ in Washington, DC and at City College of New York and compares the values to the OMI retrieved $TCNO_2$.

Both Figs. 6 and 2A show the PANDORA $TCNO_2$ retrieval with the values of the SZA plotted on the same graph showing that the direct-sun retrievals are good out to SZA = $70^O$. Depending on atmospheric conditions, retrievals using BEER's law absorption attenuation and spectral fitting for SZA > $75^O$ begin to yield non-physical values ($TCNO_2$ too small). During mid-day measurements, the signal to noise ratio is very high since over 4000 clear-sky measurements are averaged together to produce one data point every 20 seconds. Even with aerosol loading (no spectral features) or moderate cloud cover blocking the sun, the retrievals are still accurate (Herman et al., 2018).

Table 2 contains a summary of some sites that were part of short-term Discover-AQ campaigns in Maryland, Texas, California, and Colorado, two longer-term sites in South Korea, and one in New York City. Essex, Maryland is located on the Chesapeake Bay 10 km east of the center of Baltimore. The site is relatively clean (PAN = 0.3 DU) compared to the center of Baltimore (PAN = 0.45 DU), while OMI measures about the same amounts for both sites (0.28 and 0.27 DU) because the OMI FOV is larger than the distance between the two sites. The Houston Texas site contains 7 months of data from January to July 2013 with widespread $NO_2$ pollution permitting PANDORA and OMI to measure the same average values even though PANDORA observes episodes on many days when $TCNO_2$ exceeds 1.5 DU for short periods at times not observed by OMI. Observations in the small city of Fresno, California were during January when agricultural sources of $NO_2$ were at a minimum (Almaraz, 2018), but automobile traffic in the center of Fresno was significant. In this situation, PANDORA recorded the effect of automobile traffic while OMI averaged the city of Fresno and surrounding fallow agricultural areas. The Denver La Casa location is in the center of the city in an area with high amounts of local automobile traffic and near the Cherokee power generating plant. The result is a high level of average pollution (0.42 DU) while OMI measures both the city center and the surrounding relatively clean plains areas. The HUFS South Korean site is southeast of Seoul in a fairly isolated valley. However, Seoul and its surrounding areas are a widespread transported source of pollution so that both PANDORA and OMI measure elevated $TCNO_2$ amounts. In contrast, the PANDORA GIST site is on the outskirts of a small city in southwestern South Korea with significant traffic. The result is significant amounts of localized $TCNO_2$ (PANDORA = 0.42) surrounded by areas that produce little $NO_2$ leading to OMI observing a very clean 0.2 DU. The average of sites in the two tables are similar leading to ratios of PAN/OMI of 1.8 and 1.7, respectively. The estimated 50% increase in OMI retrievals of $TCNO_2$ from using the geometry-dependent reflectivity (Vasilkov, 2017) for the most polluted sites will narrow the disagreement with PANDORA. For example, OMI Seoul $TCNO_2$ may become 0.87 DU (PANDORA = 1.2 DU) and Buenos Aires 0.39 DU (PANDORA = 0.5 DU) still underestimating the amount of $NO_2$ pollution and missing the significant diurnal variation.

292        For the six sites shown, the average OMI underestimate of $TCNO_2$ is approximately a factor of 1.8 at the overpass time on a monthly average basis with occasional spikes that exceed this amount. The bias values range from 1.1 to 3.6, with higher biases tending to be associated with higher $TCNO_2$ values. The factor of 1.8 underestimate ignores the frequent large values of $TCNO_2$ at other times during the day (Fig. 7). In addition, averaging $TCNO_2$(PAN) over each entire day yields average values for the whole period that are 10 to 20% higher than just averaging over midday values that matched the OMI overpass time.  Aside from the absolute magnitude, the short-term variations (over several months) are similar for both OMI and PANDORA although mostly not correlated. If correlation coefficients $r^2$ are generated from linear fits to scatter plots of $TCNO_2$ from OMI vs PANDORA, the correlation is mostly poor (Examples, $r^2$ =:  Seoul 0.06, Mauna Loa 0.3 NASA HQ 0.7, see Figs. 4 and 5).  Additional sites with shorter PANDORA time series of $TCNO_2$ show similar behavior.

303        Duncan et al. (2016) estimated trends from OMI $TCNO_2$ time series and found that the Seoul metropolitan area had a decrease of -1.5 ± 1.3 %/Year (2005 – 2014) consistent with OMI estimated change of –1.4 ± 1%/year (2012 -2018) in this paper.  However, for the small area near Yonsei University, the decrease estimated from PANDORA is -5.8 ± 0.75 %/Year. Park (2019) estimates that metropolitan Seoul has decreased in population even as surrounding areas have increased population.

308        The average percent differences between OMI and PANDORA shown in Fig. 7 are relatively constant over time for each site with small changes over each multi-year observation period. The differences between OMI and PANDORA are provided by forming the percent differences of the daily $TCNO_2$ values (Fig. 7) in the form 100(OMI – PAN)/PAN. Also shown are the average percent differences and the linear fit slopes in percent change per year of the percent differences over the multi-year period. For example, the Boulder percent difference goes from -31% to -23% over 4 years. Of the six sites in shown in Fig. 7, two have statistically significant slopes, Seoul South Korea 2.1±0.5 %/Year and NASA Headquarters in Washington DC 3.4±0.9 %/Year at the 2σ level suggesting a significant area average increase in pollution compared to PANDORA's local values.

317        For some sites (see Fig. 7), PANDORA and OMI trends are the same within statistical uncertainty (Waterflow, NM, Buenos Aires, and Mauna Loa) while the other 3 sites show significantly different trends (Boulder, NASA HQ, and Seoul).

320        The results for Busan (from Fig. 3) show a least squares average for the percent difference of -48 ± 0.8% for the 2012 – 2018 period with a slope of 6.8 ± 1%/Year. There is a decrease in the percent difference after October 2015 (Fig. 3) that is mainly from PANDORA seeing less $TCNO_2$ than during the 2012 – 2014 period. There is a gap in the Busan time series from July 2014 until April 2015 when the original PANDORA was replaced with a new instrument. The calibrations of both PANDORAS appear to be correct. Because of the break in the time series it is not clear whether there was a change in local conditions around Pusan University compared to the wide area observed by OMI.

## 3.1 Diurnal Variation of TCNO$_2$ Compared to OMI Retrievals

Figure 8 shows details of the daily diurnal variation of TCNO$_2$ on the roof of NASA Headquarters Washington, DC adjacent to a major cross-town highway (I695) for every day during each month of 2015 for local time vs DOY. The midday observing local standard time for OMI is marked for each graph. Displaying an entire year of daily (2-minute time resolution) PANDORA data shows that the high values of TCNO$_2$ are a frequent occurrence but do not occur every day.

The amount of TCNO$_2$ is mostly from the adjacent highway and the surrounding urban area with heavy traffic. The relatively moderate TCNO$_2$ values (0.4 to 0.8 DU) are probably a testament to the effectiveness of catalytic converters mandatory on all US automobiles in such a high traffic area (Bishop and Steadman, 2015). The same data are plotted in Fig. 6 for 8 June 2017 showing that OMI reasonably matched the values seen by PANDORA at 14:00 and 15:00 but was not available to observe high values that occurred in the morning.

Figure 9 contains the daily TCNO$_2$ diurnal variability vs DOY for each month measured by a PANDORA from the roof of a building on the CCNY (City College of New York) campus in the middle of Manhattan in New York City (NYC). From the values shown, the pollution levels are quite high, rivaling the pollution levels in Seoul, South Korea (see Fig. 5). OMI at its mid-day overpass time would detect some of the high-level pollution events, but miss many others occurring mostly in the afternoon. There are a significant number of days in all the months where the TCNO$_2$ levels appear to be low (e.g., blue color in July and October), but the blue color still represents significant pollution levels (TCNO$_2$(PAN) > 0.5 DU) that are small only compared to the peak values during the month (TCNO$_2$(PAN) > 1 DU). The highest amount of TCNO$_2$ recorded during 2018 was about 5DU on 13 July 2018 from 11:20 and 12:30 EST (a time with very light winds (1 km/hr) and moderate temperature (25$^O$C). There were many smaller peaks between 2 and 3 DU throughout the year. Extreme cases of high NO$_2$ amounts are frequently associated with the local meteorology indications of stagnant air (Harkey et al., 2015), The same data are shown in Fig. 6 for two days, 7 May 2018 and 7 June 2018 showing the comparison with OMI and the occurrence of much higher values of TCNO$_2$ in the morning and afternoon.

For both Washington DC (Fig. 8) and New York City (Fig. 9) there is strong day-to-day and month to month variability that depends on the local meteorological conditions (Seo et al., 2018; Zeng et al., 2015) and the amount of automobile traffic in the area (Andersen et al., 2011; Amin et al., 2017). High TCNO$_2$ events occur most often in the afternoon such that the OMI overpass near 13:30 would miss most high TCNO$_2$ events. Poor air quality affecting respiratory health would be improperly characterized by both the OMI average values being too low (Fig. 4) and by missing the extreme pollution events that occur frequently in the late afternoon. The high value of TCNO$_2$ that occurred on 5 August (2.2 DU) at 07:45 EST for Washington DC is not a retrieval error (SZA less than 70$^O$), but is a one-time anomaly in 2015 compared to more usual high values of 1.5 DU with an occasional spike to 2 DU. It should be noted that TCNO$_2$ does not accurately represent the NO$_2$ concentration at the surface, since it is mostly a measure of the amount in the lower 2 km. However, it is roughly proportional to the surface measurements close to the pollution sources (Bechle et al., 2013; Knepp et al., 2014) with the exact proportionality dependent on the profile shape near the ground.

Similar daily diurnal variation graphs of TCNO$_2$ (Figs. 8 and 9) could be shown for each site.
However, the basic idea is the same for each site. OMI underestimates the amount of TCNO$_2$ because of
its large FOV and misses most of the peak events at other times of the day. For some sites, such as Busan
and Seoul, the peak values can reach 3 DU and above late in the afternoon, which are never seen by OMI
(Herman et al., 2018).
Figure 10 for CCNY is similar to the graphs in Figs. 4 – 6 showing the relative behavior between
PANDORA and OMI but including only OMI pixels that are at a distance D < 30 km from CCNY. The results
are almost identical to those when D < 80 km. There is a period in March 2018 when OMI TCNO$_2$ slightly
exceeded that measured by PANDORA. OMI with its large FOV may be seeing part of the chemically driven
seasonal variation, while PANDORA is seeing a nearly constant source driven amount mostly from
automobile traffic. For most days during 2018, PAN(TCNO$_2$) > OMI(TCNO$_2$) with the average value for PAN
= 0.65 DU and for OMI = 0.45 DU (Fig. 10 Panel B). The percent difference plot shows that there is a
systematic increase between PANDORA and OMI TCNO$_2$ from a value 10% to a value of 50%.

## 4.0 Summary

Examination of long-term TCNO$_2$ monthly average time series from OMI satellite and PANDORA
ground-based observations show that OMI systematically underestimates the amount of NO$_2$ in the
atmosphere by an average factor of 1.5 to 2 at the local OMI overpass time near the equator crossing time
of 13:30±1:30. As shown in Fig. 7 for TCNO$_2$, 100(OMI – PAN)/PAN least squares mean underestimates are
much larger than error estimates. These differences are reduced for the smaller pixel size TropOMI TCNO2
values (Judd et al.,2019).  In addition, the PANDORA diurnal time series for every day during a year at each
site (only two typical sites are shown in this paper, NYC and NASA-HQ) shows peaks in TCNO$_2$ that are
completely missed by only observing at mid-day (see Figs. 6, 8, 9, and A2).  The result is that estimates of
air quality related to health effects from OMI observations are strongly underestimated almost
everywhere as shown at all the sites with a long PANDORA record. In comparisons to PANDORA, OMI data
are mostly uncorrelated or weakly correlated (e.g., Seoul correlation coefficient r$^2$ = 0.06, Mauna Loa r$^2$ =
0.3), while NASA HQ in Washington, DC shows a correlation on a seasonal basis (NASA HQ r$^2$ = 0.7)
suggesting a wide area coordinated source of NO$_2$ (most likely automobile traffic). The data from CCNY
shows some correlation between the locations of the peaks and troughs. Seven short term TCNO$_2$ time
series were examined showing similar results (Table 1), except when the pollution region is widespread
as in the Seoul South Korea region. The conclusion is that while OMI satellite TCNO$_2$ data are uniquely
able to assess regional long-term trends in TCNO$_2$ and provide a measure of the regional distribution of
pollutants, the OMI data cannot properly assess local air quality or the effect on human health over
extended periods in urban or industrial areas. This will continue to be the case, but to a lesser degree,
when the OMI TCNO$_2$ data are improved by reprocessing with a new geometry-dependent reflectivity
(Vasilkov, 2017) and by the smaller FOV of TropOMI. The analysis shows that locating PANDORAs at
polluted sites could provide quantitative corrections for spatial and temporal biases that affect the
determination of local air quality from satellite data. Satellite detection of diurnal variation of TCNO$_2$ will
be improved with the upcoming launch of three planned geostationary satellites over Korea, US, and
Europe To verify the proper operation of the various PANDORA instruments, a similar analysis for Total
Column Ozone TCO was performed (see Appendix) and shows close agreement between OMI and
PANDORA, with the largest difference occurring for Mauna Loa Observatory at 3.4 km altitude, where
PANDORA misses the ozone between the surface and 3.4 km.
**Appendix**
**Ozone:** This section shows the corresponding PANDORA total column ozone (TCO) values
compared to OMI TCO for Busan South Korea (Fig. A1) that shows close agreement for the entire 2012 –
2017 period. The different fields of view for OMI and PANDORA have a much smaller effect because of
the greater spatial uniformity of stratospheric ozone compared to tropospheric $NO_2$. Additional sites are
summarized in Table A1. The largest TCO difference (15 DU or 5.6%) occurs for Mauna Loa Observatory
(Altitude = 3.4 km) compared to OMI (Average altitude = Sea Level). The close results show that the
PANDORA was working properly and pointing accurately at the sun. The PANDORA TCO data shown here
use a mid-latitude effective ozone temperature correction from model calculations that may not be
accurate of each individual site (Herman et al., 2017).  The ozone retrievals shown here use an average
effective ozone temperature instead of a locally measured ozone temperature (Herman et al.,
422 2015;2017).



Table A1  Average values of $TCO_3$ for PANDORA  and OMI

| Location | PAN (DU) | OMI (DU) | Percent Difference |
|---|---|---|---|
| Mauna Loa Observatory Hawaii (3.394 km)* | 254 | 269 | 5.6 |
| NASA HQ Washington DC (0.02 km) | 308 | 314 | 1.9 |
| Waterflow New Mexico (1.64 km) | 293 | 292 | 0.3 |
| Yonsei University Seoul South Korea (0.07 km) | 317 | 325 | 2.5 |
| Busan University Busan South Korea(0.03 km) | 313 | 315 | 0.6 |
| Boulder, Colorado (NOAA Bldg) (1.617 km) | 299 | 302 | 1.0 |
| Buenos Aires, Argentina (0.025 km) | 279 | 284 | 1.8 |
| Essex, Maryland (0.012 km) | 299 | 301 | 0.7 |
| Baltimore, Maryland  (0.01 km) | 296 | 296 | 0.0 |
| Fresno, California (0.939 km) | 306 | 309 | 1.0 |
| Denver La Casa Colorado (1.6 km) | 292 | 294 | 0.7 |
| Gwangju Institute of Science and Technology (GIST) S. Korea (0.021 km) | 302 | 307 | 1.6 |
| Hankuk University Foreign Studies (HUFS ) South Korea (0.04 km) | 318 | 326 | 2.5 |
| City College Manhattan New York City (0.04 km) | 316 | 325 | 2.8 |
| **Average** | **299** | **304** | **1.6** |


**Mauna Loa TCNO₂**: Figure A2 shows the diurnal variation of TCNO2 at MLO on specific days
3,4,7, and 8 of June 2016 along with the variation in SZA. This shows that the MLO is polluted by $NO_2$
with column amounts in excess of stratospheric amounts (approximately 0.1 DU) even though there are
no local sources. OMI retrievals of TCNO2 on each day are much lower (about 0.12 DU) because of the
averaging over OMI's large FOV that includes very clean ocean areas.

**Acknowledgement:** This project is supported by the Korea Ministry of Environment (MOE) as Public Technology
Program based on Environmental Policy (2017000160001), by the Los Alamos National Laboratory's Laboratory
Directed Research and Development program  and by the NASA Pandora project managed by Dr. Robert
Swap.

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

Fig. 2. Monthly average values of $TCNO_2$ for OMI and PANDORA at OMI overpass times
Fig. 3 Extended time series for Busan. Left Panel: individual matching PANDORA and OMI data
points for the overpass time ± 6 minutes. Right Panel: monthly averages.
Fig. 4. PANDORA compared to OMI. Extended $TCNO_2$ overpass time series for Mauna Loa
Observatory, Hawaii, NASA Headquarters, Washington DC, and Waterflow, New Mexico.
Fig. 5. PANDORA compared to OMI. Extended $TCNO_2$ overpass time series for Seoul South Korea,
Boulder, Colorado, and Buenos Aires, Argentina (Raponi et al. 2018).
Fig.6 Diurnal variation of $TCNO_2$ on a single day 1) Two km north of Waterflow, NM near a
power plant, 2) On the roof of NASA Headquarters Washington, DC and 3) On the roof or a
building at CCNY City College of New York, New York City
Fig. 7 Percent differences between OMI and PANDORA. The slopes are the absolute change in the
percent difference. For example, the Boulder percent difference goes from -31% to -23% over 4 years.
The LS Means are least squares means with the corresponding error estimates
Fig. 8A $TCNO_2$ diurnal variation (DU) from January to June, NASA Headquarters Washington, DC
from January 2015 to June 2015.  The approximate OMI overpass time near 13:30 hours is marked.
Fig. 8B $TCNO_2$ diurnal variation (DU) from July to December, NASA Headquarters Washington, DC from
July 2015 to December 2015.  The approximate OMI overpass time near 13:30 hours is marked
Fig. 9A $TCNO_2$ diurnal variation (DU) at CCNY in New York City January to June 2018. The approximate
OMI overpass time near 13:30 hours is marked.
Fig. 9B $TCNO_2$ diurnal variation at CCNY in New York City July to December 2018. The peak near 5 DU
occurs on 13 July 2018 between 11:20 and 12:30 EST. The approximate OMI overpass time near 13:30
hours is marked.
Fig. 10 $TCNO_2$ overpass time series for CCNY in Manhattan, New York City. Panel A: OMI
overpass $TCNO_2$ (Black) compare with OMI (Red). Panel B: Monthly Lowess(0.08) fit to the daily
overpass data. Panel C: Percent difference 100(OMI – PAN)/PAN calculated from the data in
Panel A
Fig. A1 Monthly average values of TCO for OMI and PANDORA at OMI overpass times for Busan South
Korea. Shaded area represents the KORUS-AQ campaign period.
Fig. A2. The diurnal variation of TCNO$_2$ at MLO on 4 days during June 2016 compared to OMI TCNO$_2$
(small square). Shaded areas represent high SZA conditions where the PANDORA retrievals are not
accurate.

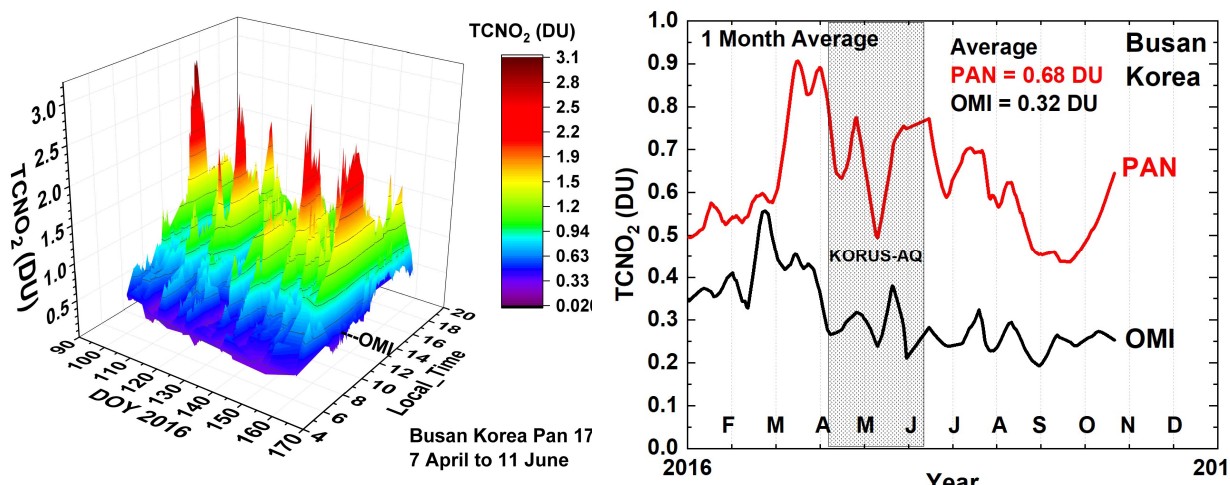

Fig 1 Diurnal variation of TCNO₂ measured at Pusan University in Busan South Korea

Fig. 2. Monthly average values of TCNO₂ for OMI and PANDORA at OMI overpass times




**FIGURE 1**                                        **FIGURE 2**



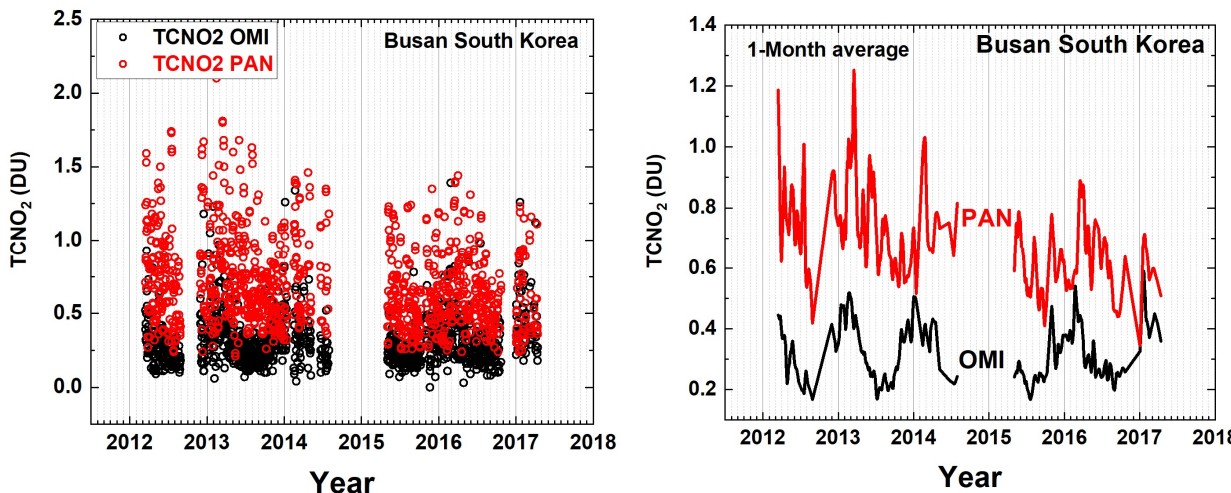

Fig. 3 Extended time series for Busan. Left Panel: individual matching PANDORA and OMI data points for the overpass time ± 6 minutes. Right Panel: monthly averages.



**FIGURE 3**

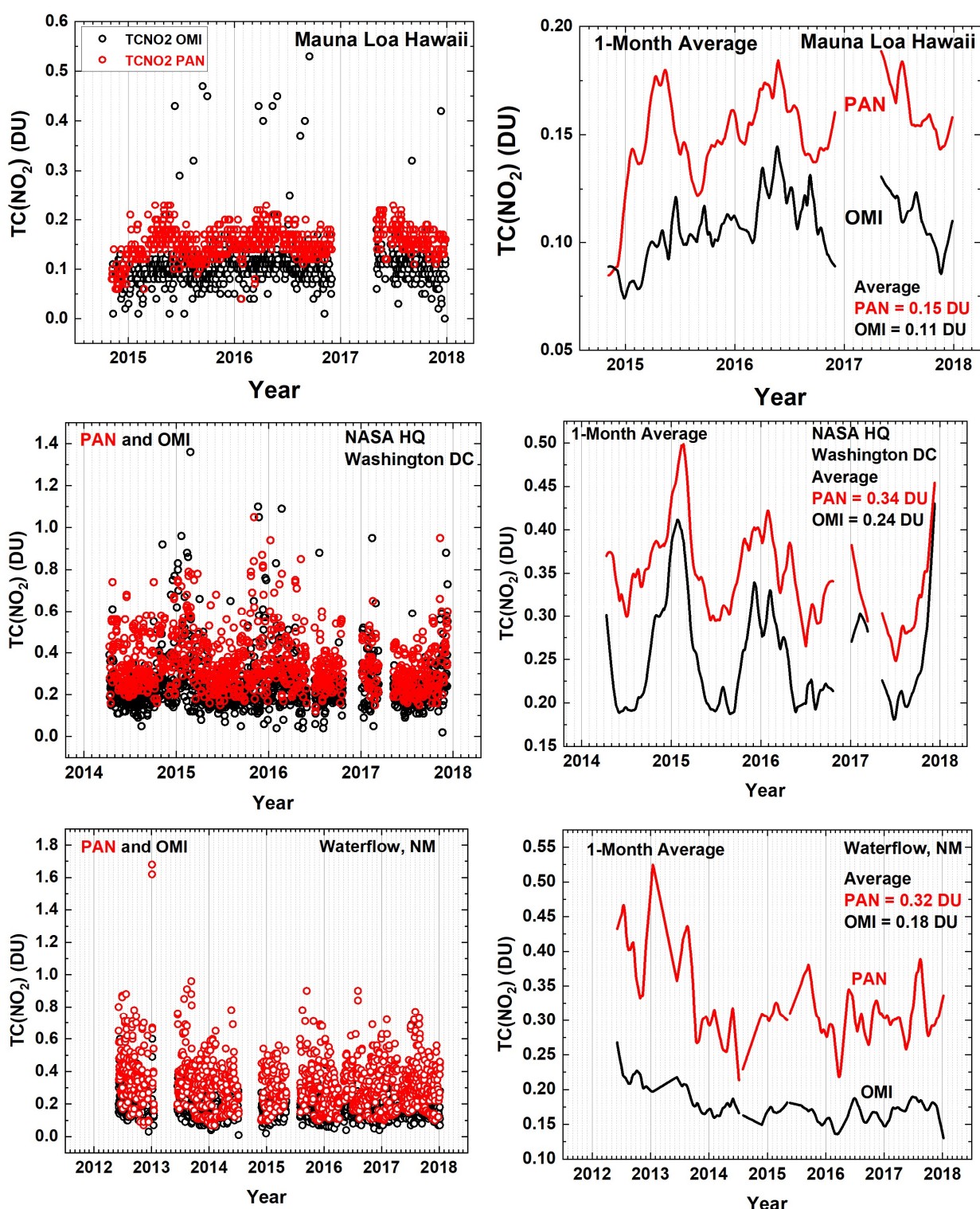

Fig. 4. PANDORA compared to OMI. Extended TCNO2 overpass time series for Mauna Loa Observatory, Hawaii, NASA Headquarters, Washington DC, and Waterflow, New Mexico.

**FIGURE 4**

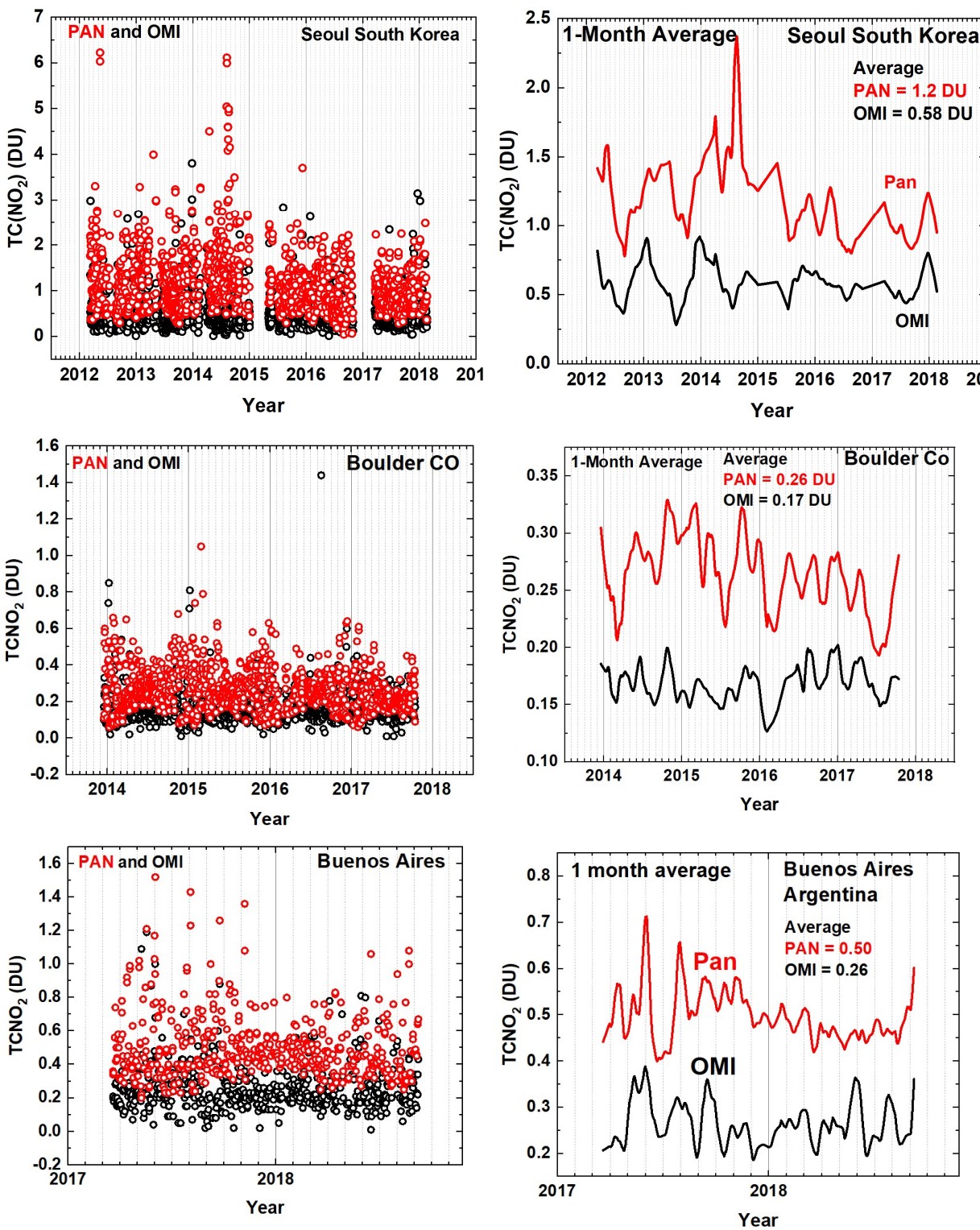

Fig. 5.  PANDORA compared to OMI. Extended TCNO₂ overpass time series for Seoul South Korea, Boulder, Colorado, and Buenos Aires, Argentina (Raponi et al. 2017).

**FIGURE 5**

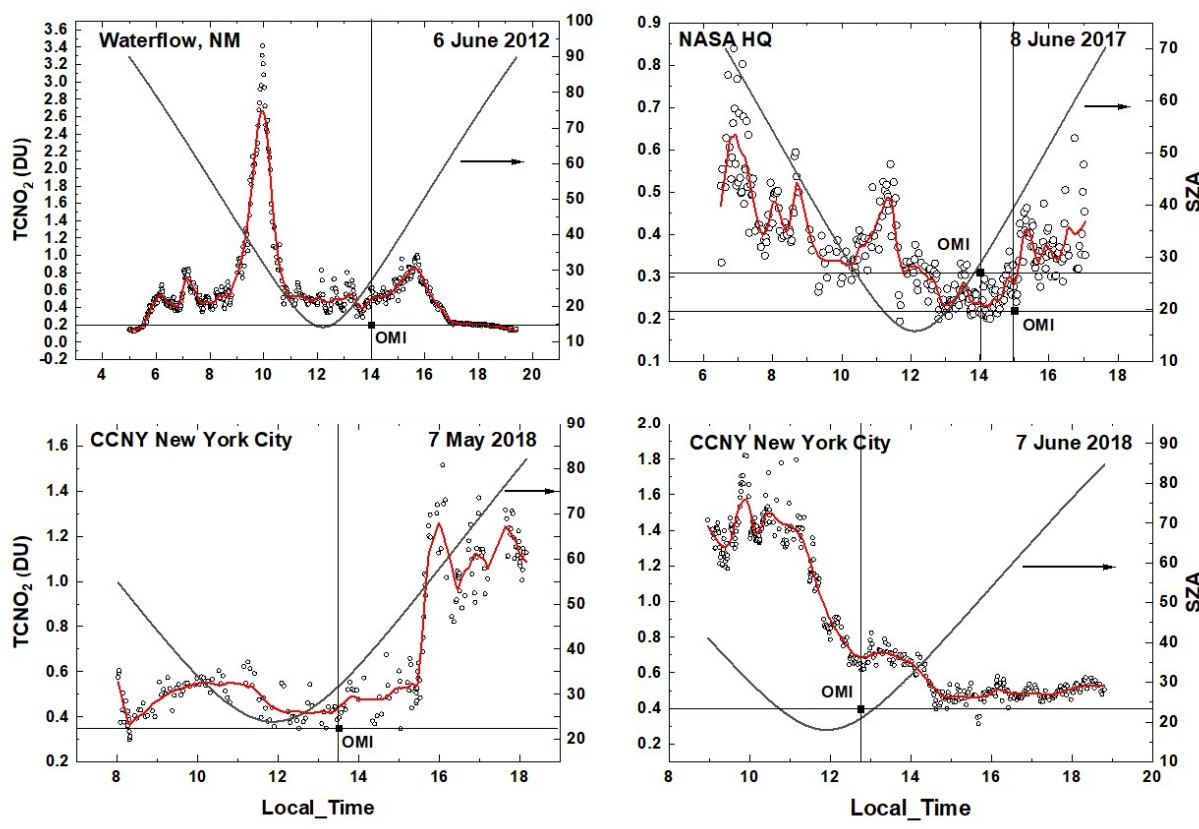

Fig.6 Diurnal variation of TCNO$_2$ on a single day 1) Two km north of Waterflow, NM near a power plant, 2) On the roof of NASA Headquarters Washington, DC and 3) On the roof or a building at City College of New York, New York City






**FIGURE 6**





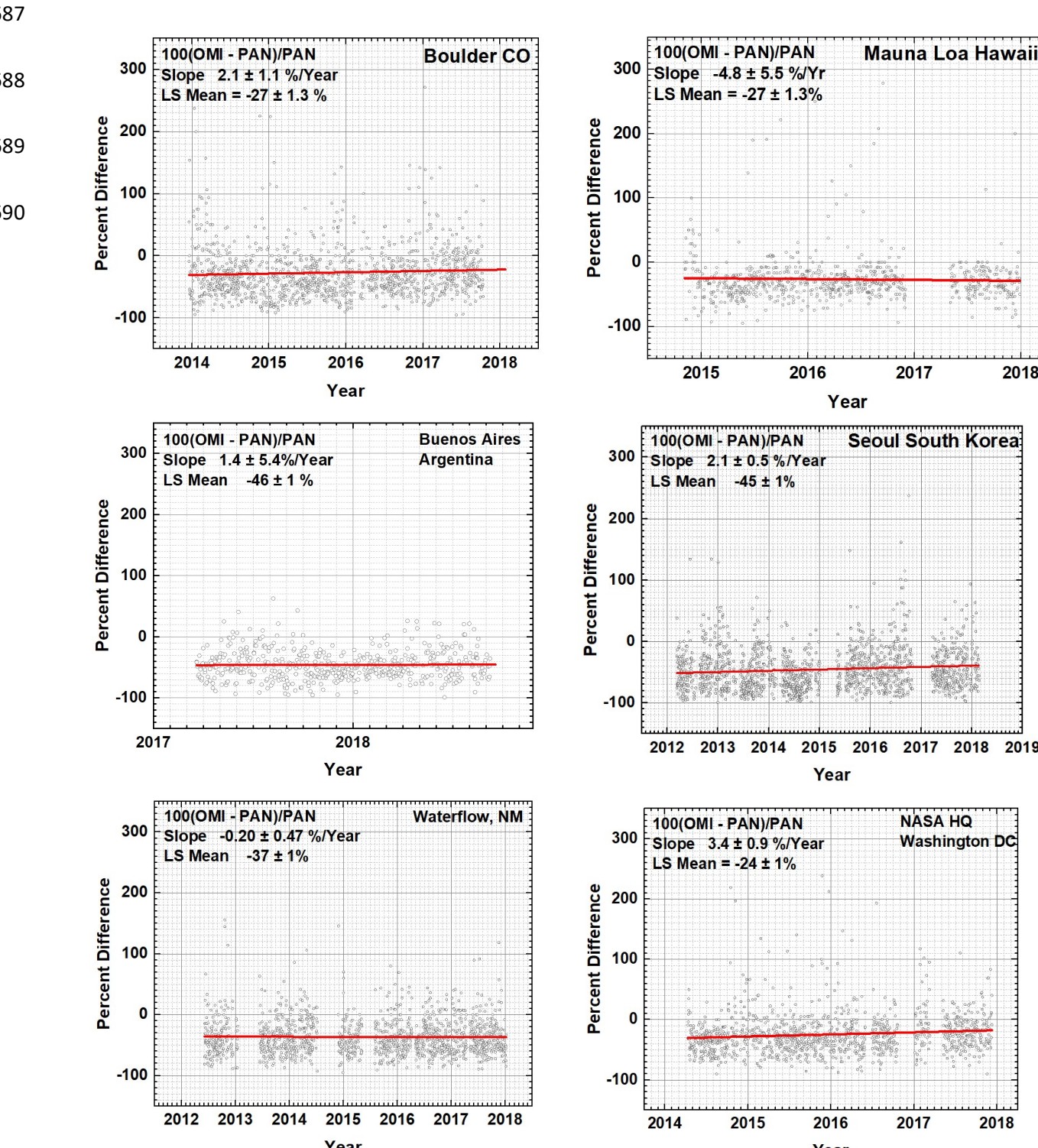

Fig. 7 Percent differences between OMI and PANDORA. The slopes are the absolute change in the percent difference. For example, the Boulder percent difference goes from -31% to -23% over 4 years. The LS Means are least squares means with the corresponding error estimates

**FIGURE 7**


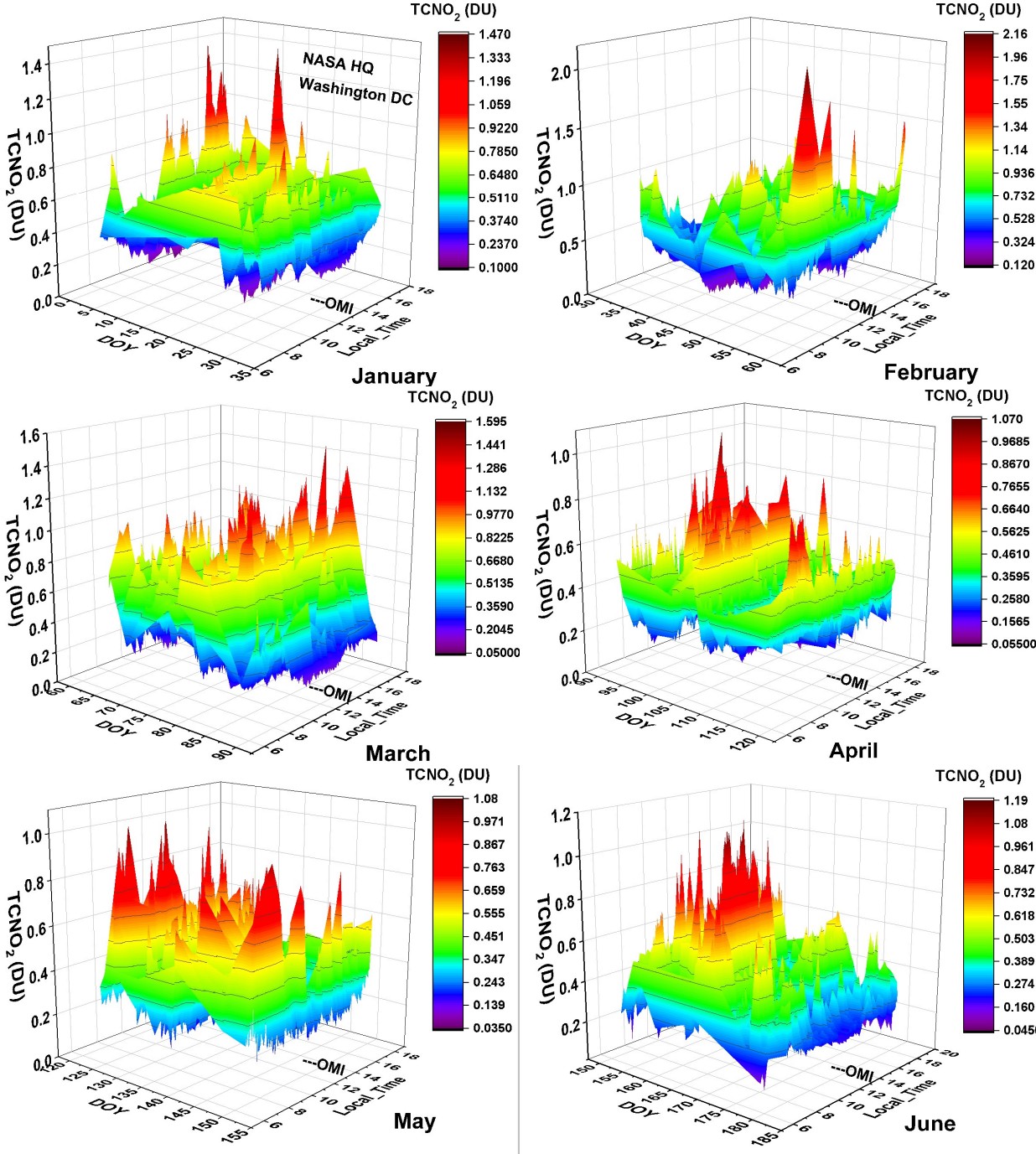

Fig. 8A TCNO$_2$ diurnal variation (DU) from January to June, NASA Headquarters Washington, DC from January 2015 to June 2015. The approximate OMI overpass time near 13:30 hours is marked


**FIGURE 8A**

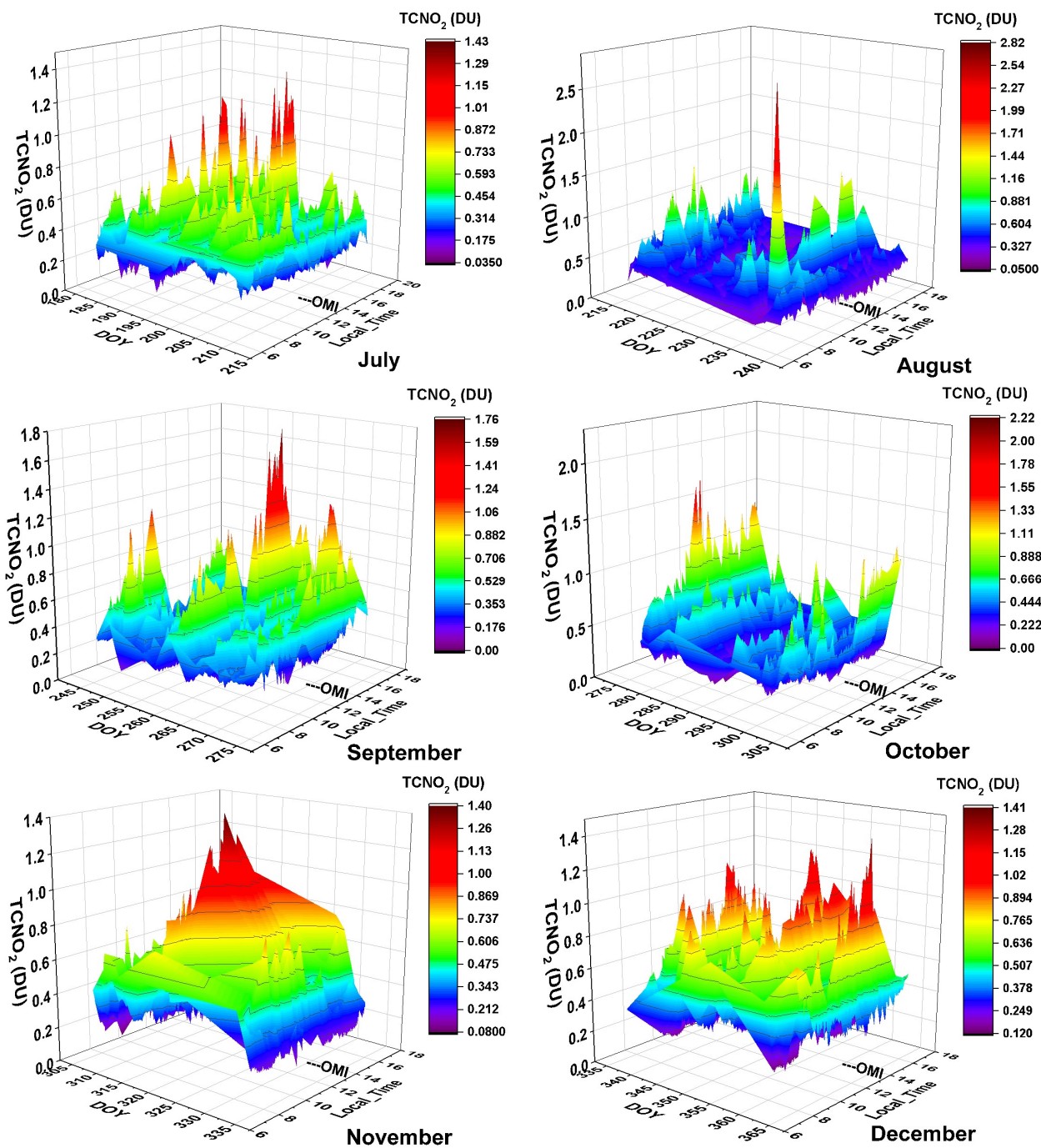

Fig. 8B TCNO₂ diurnal variation (DU) from July to December, NASA Headquarters Washington, DC from July 2015 to December 2015.  The approximate OMI overpass time near 13:30 hours is marked.


**FIGURE 8B**



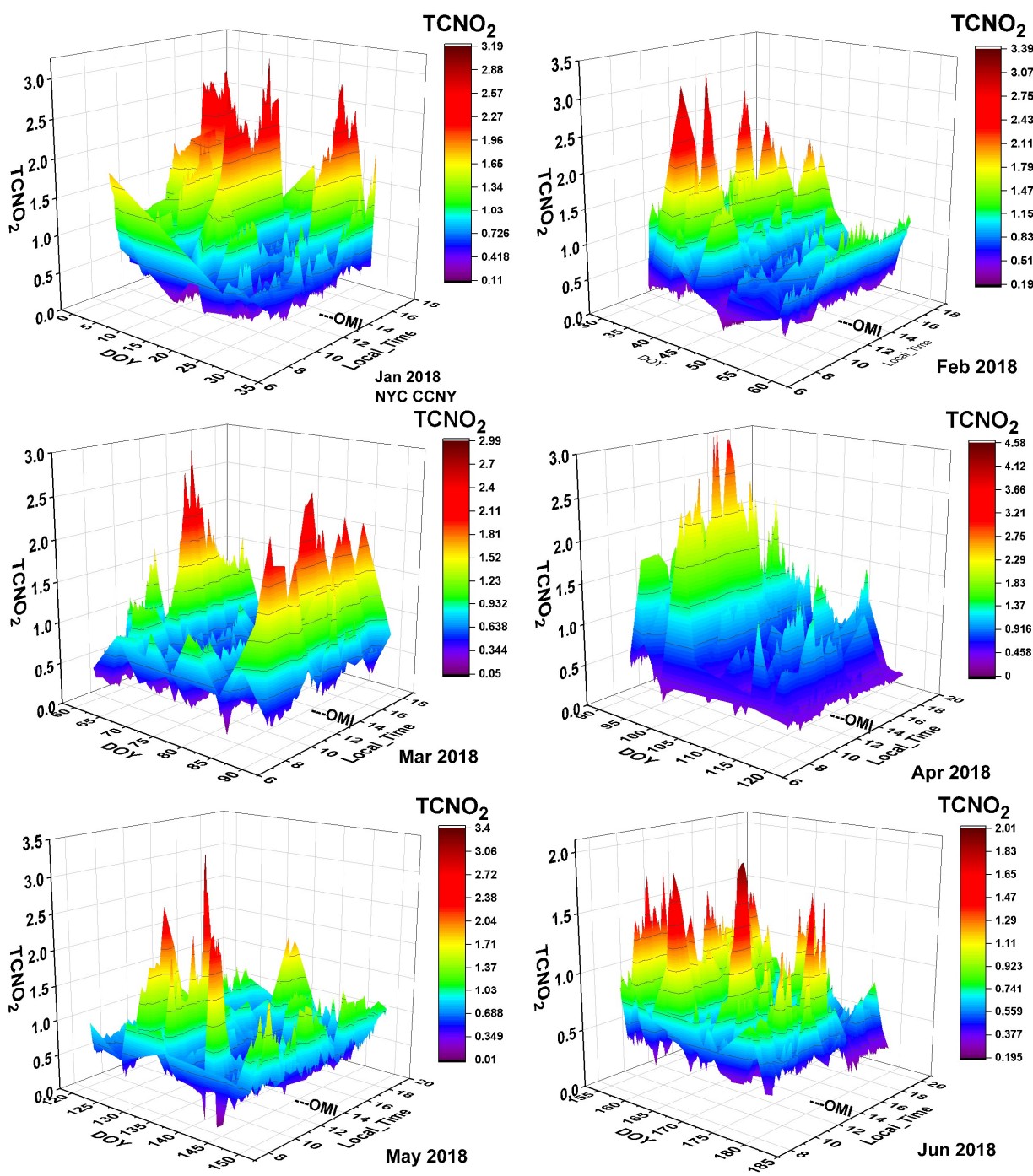

Fig. 9A TCNO₂ diurnal variation (DU) at CCNY in New York City January to June 2018. The approximate OMI overpass time near 13:30 hours is marked


**Figure 9A**

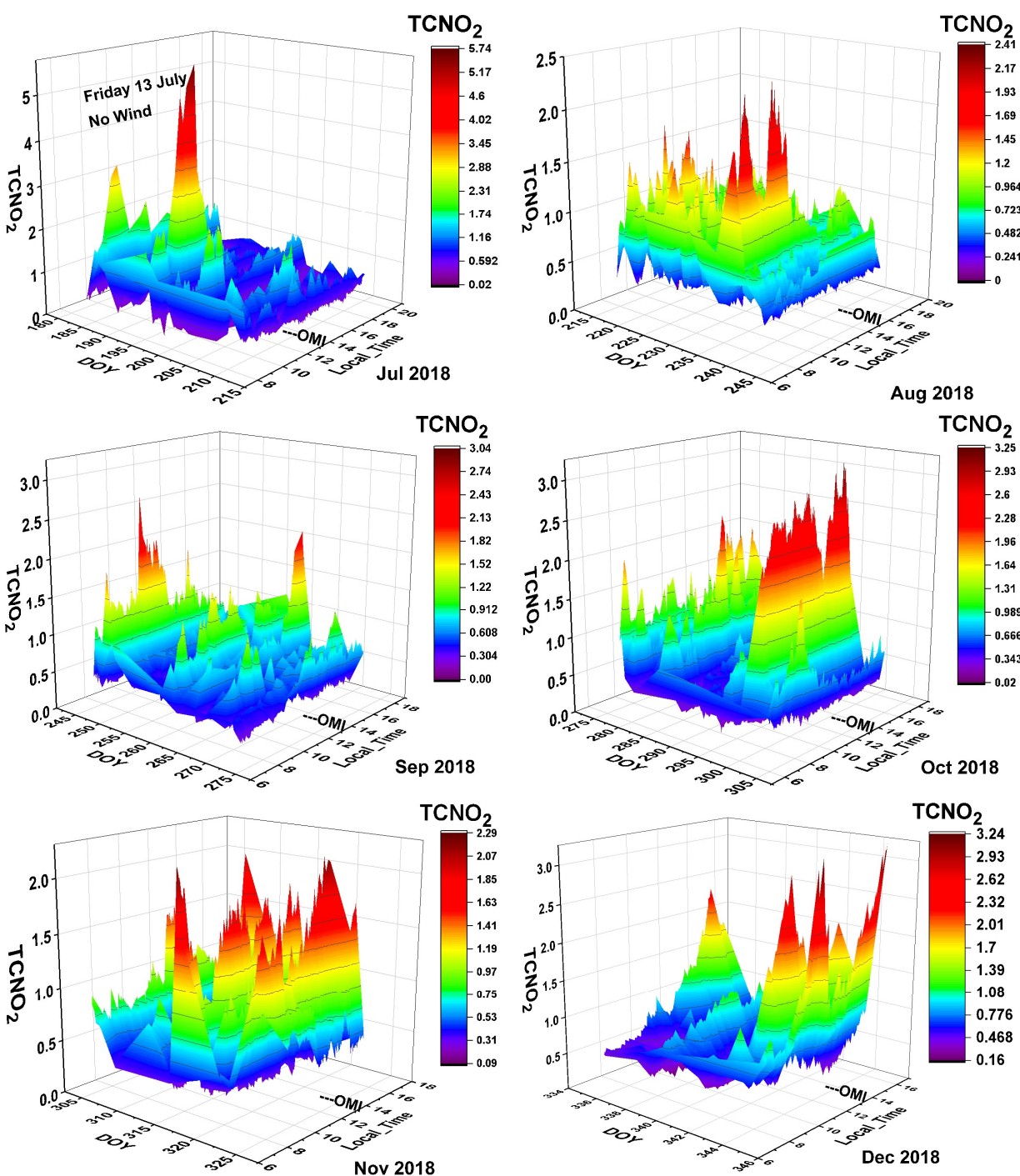

Fig. 9B TCNO$_2$ diurnal variation (DU) at CCNY in New York City July to December 2018. The peak near 5 DU occurs on 13 July 2018 between 11:20 and 12:30 EST. The approximate OMI overpass time near 13:30 hours is marked.


**Figure 9B**

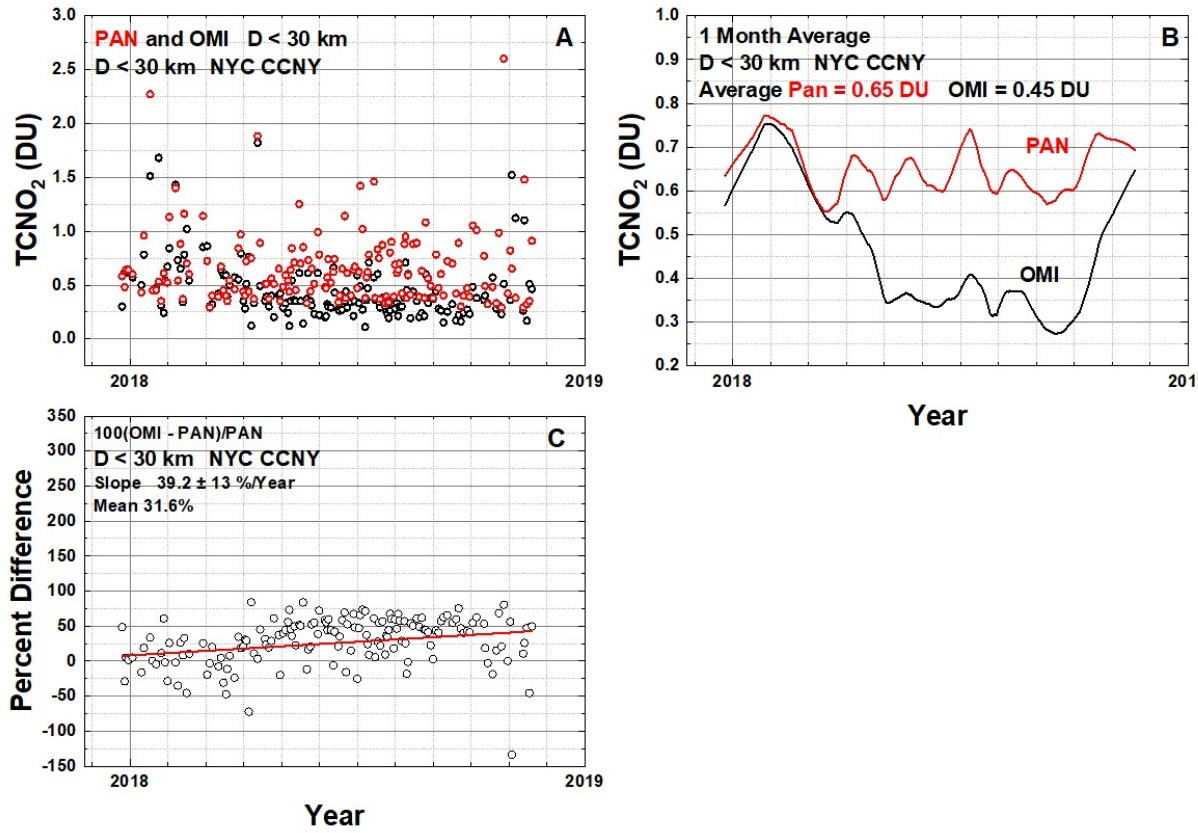

Fig. 10 TCNO$_2$ overpass time series for CCNY in Manhattan, New York City. OMI pixels are at a distance D < 30 km from CCNY. Panel A: OMI overpass TCNO$_2$ (Black) compare with OMI (Red). Panel B: Monthly Lowess(f) fit to the daily overpass data. Panel C: Percent difference 100(OMI – PAN)/PAN calculated from the data in Panel A


**Figure 10**

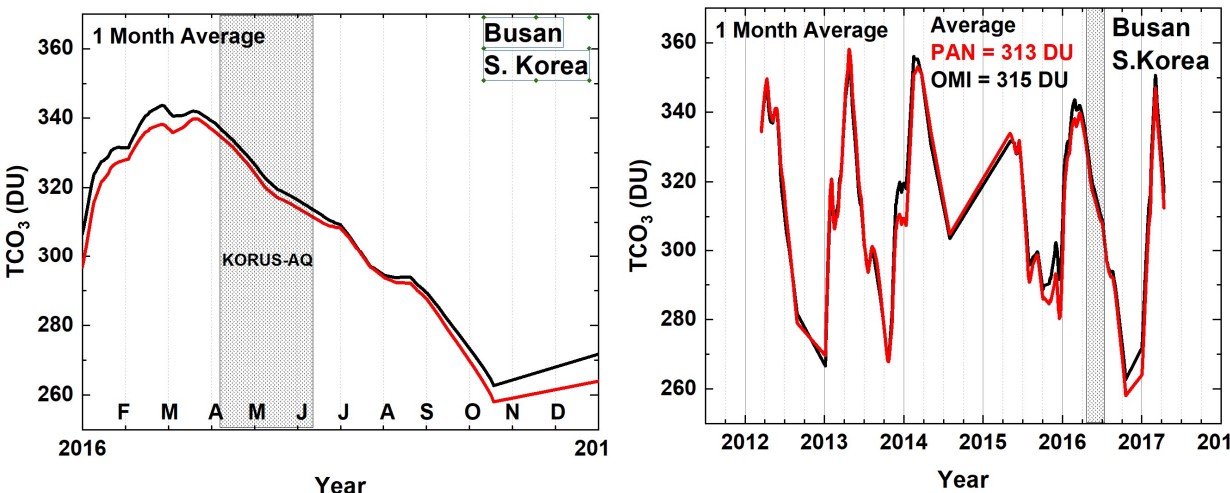

Fig. A1 Monthly average values of TCO$_3$ for OMI and PANDORA at OMI overpass times for Busan South Korea. Shaded area represents the KORUS-AQ campaign period.



**FIGURE A1**

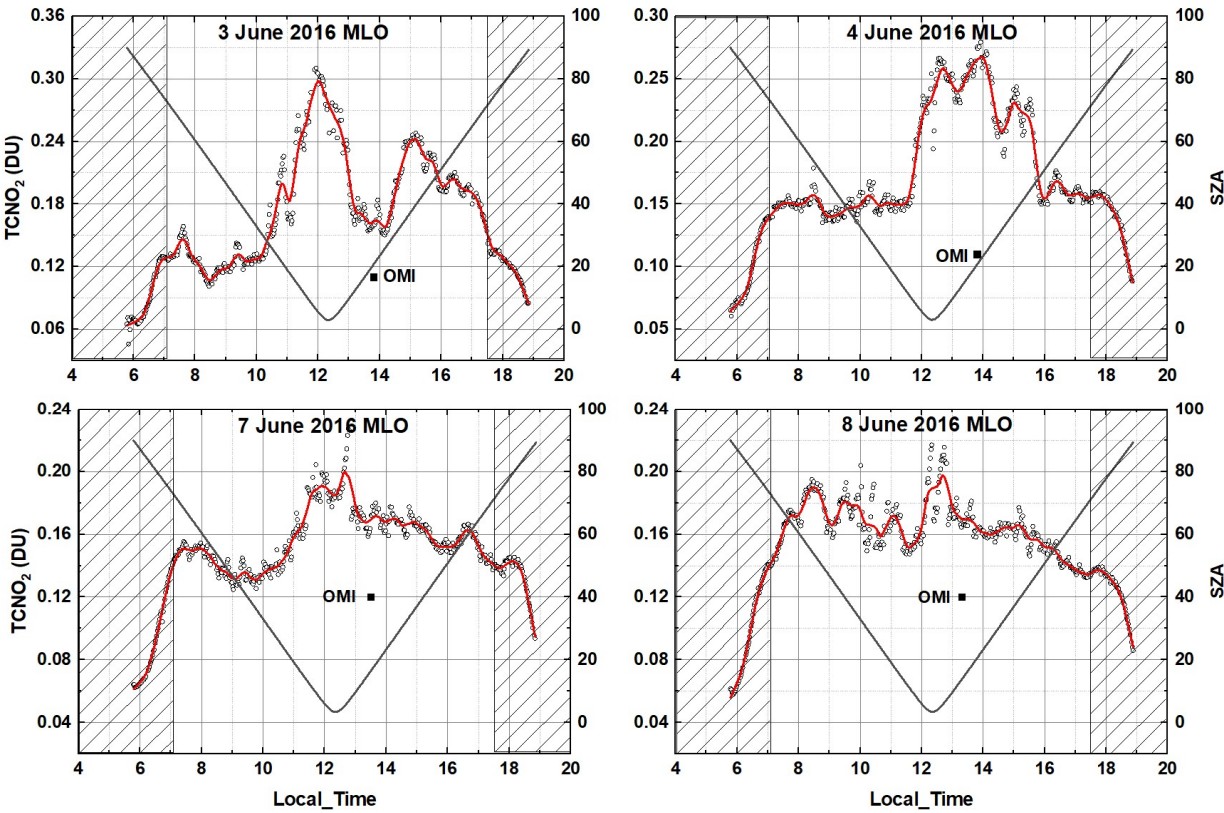

Fig. A2. The diurnal variation of $TCNO_2$ at MLO on 4 days during June 2016 compared to OMI $TCNO_2$ (small square). Shaded areas represent high SZA conditions where the PANDORA retrievals are not accurate.







**FIGURE A2**


**Author Contributions**: Jay Herman produced all of the figures and text, Jhoon and Jae Kim contributed
the Korean PANDORA data, Manvendra Dubey contributed the data from Waterflow, New Mexico and
the $CO_2$ analysis, Marcel Raponi contributed the Buenos Aires PANDORA data, And Maria Tzortziou
contributed the PANDORA data from CCNY.