# Peer review of "Underestimation of Column NO2 Amounts from the OMI Satellite Compared to Diurnally Varying Ground-Based Retrievals from Multiple Pandora Spectrometer Instruments"

_Atmospheric Measurement Techniques, 2019_

## Referee Comment (RC1) · Anonymous Referee #1 · 31 May 2019

General comments

The manuscript presents an evaluation of the OMI NO2 columns against ground-based observations at different sites using Pandora measurements. The authors find that OMI underestimates as expected the GB measurements and they attribute this underestimation to retrieval issues and differences in field of view. They also discuss the effect of NO2 daily cycle. The results are a good addition to the existing literature but their presentation and the way they reach the conclusions might be improved quite a bit as I suggest below.

[Figure]

Specific comments

1. Abstract: L8-15 Should this description of the sites be here in the abstract? Maybe you could write in a more concise way and focus on the results here instead...

2. L33 Perhaps a reference here, e.g. Krotkov et al. (2016) Krotkov, N. A., McLinden, C. A., Li, C., Lamsal, L. N., Celarier, E. A., Marchenko, S. V., Swartz, W. H., Bucsela, E. J., Joiner, J., Duncan, B. N., Boersma, K. F., Veefkind, J. P., Levelt, P. F., Fioletov, V. E., Dickerson, R. R., He, H., Lu, Z., and Streets, D. G.: Aura OMI observations of regional SO2 and NO2 pollution changes from 2005 to 2015, Atmos. Chem. Phys., 16, 4605-4629, https://doi.org/10.5194/acp-16-4605-2016, 2016.

3. L36 you mean "OMI TCNO2 underestimation"?

4. L42 Maybe you can rewrite this more specifically e.g. mentioning that OMI does not capture higher values occurring after the overpass time and thus cannot be used alone for estimating the hazard related to bad AQ.

5. L116-117 Are these overpass files based on the minimum distance between pixel center and the GB site? There is also the possibility to use the pixel actually including the GB site; this might be not the same than the one with the minimum distance from the pixel center. Did you check that? You might also want to analyse the large pixels separately (the ones on the sides of the swath are significantly larger than in nadir) and see if the underestimation is actually related to the size of the pixel and how.

6. Fig. 3 first panel: Because of the long time period this plot is really busy and doesn't add much to the one with the monthly data on the right side: maybe you could think to replace it with a scatterplot instead? Same for fig. 4 and 5.

7, L245-249 and L261-267 There seems to be a repetition here

8. Fig. 6 Could a similar picture be done for the rel.dif. as a function of the OMI pixels size? This might help supporting your conclusion that the underestimation is due to the large FOV of OMI. (see also point 5)

[Figure]

9. Fig. 6 and 9 Can you explain why do you expect from these trend plots? Why do you think the rel. difference should change?

10. Fig. 7-8 These 3D plots are maybe not so clear if you want to compare the daily cycles in different months: maybe you could replace them with a pcolor or contour -type of plots or even better adding a 1D plot with the mean daily cycles for each month. You could be able to better visualise seasonal differences in the daily cycle. Again, about the daily cycle, you could compare your results with this paper by Boersma et al. 2009, where the seasonal changes in the NO2 daily cycle were analysed in details.

https://www.atmos-chem-phys.net/9/3867/2009/acp-9-3867-2009.pdf

11. Summary: You could add a couple of sentences on the potential of the new retrievals from TROPOMI (much smaller pixel) as well as TEMPO higher (hourly) temporal resolution.

Technical comments

L30 foe -> for

L169 PANDRA -> PANDORA

L209-211 This is a bit of a repetition

---

## Referee Comment (RC2) · Anonymous Referee #2 · 11 Jun 2019

**Review of "Underestimation of Column NO2 Amounts from the OMI Satellite Compared to Diurnally Varying Ground-Based Retrievals from Multiple Pandora Spectrometer Instruments" by Jay Herman et al**

This paper presents a comparison between OMI NASA v3.1 and PANDORA total NO2 VCD, showing a clear under-estimation of the OMI data at 7 long-term sites and 6 campaign-based sites. The results at most of the sites are presented and discussed and few arguments for the general under-estimation result are mentioned. Although the paper is interesting and fulfill the scope of AMT, there is a lack of reference to literature (previous similar studies and scientific proof/reference of why such differences at the different sites). Sensitivity tests or further comparisons on OMI pixel sizes (edge and center of the swath, different position of the pixels, GB time-selection) could be done to help justifying the proposed conclusion. I recommend the publication after the suggested revisions.

**General comments:**

The paper is short and easy to read, but it lack some "proof" of the proposed explanation of the OMI under-estimation (argument 1= "Because of the local inhomogeneity of NO2 emissions, the large OMI FOV is the most likely factor when comparing OMI TCNO2 to retrievals from the small PANDORA effective FOV", line 20 and argument 2= "OMI estimated air mass factor, surface reflectivity, and the OMI 24x13 km2 FOV (field of view) are three factors that can cause OMI to underestimate TCNO2", line 18).

Some sensitivity tests on the how much the choices made for the OMI pixel selection (FOV distance d<5km for an SZA<70, line 165) and PANDORA selection ("daily data matched to the OMI overpass times ±6miutes", line 87) affect the results would support argument 1 (or at least give an uncertainty range). Additional comparison (or at least further comments on other OMI retrievals, such as DOMINO (Boersma et al., 2011) or QA4ECV (Boersma et al., 2018)), would support argument 2.

Moreover, a lot of (redundant) figures are given (daily and monthly panels in Fig 3, 4, 5 and 9) could be simplified by plotting the mean and the variability – or a scatter plot of OMI vs PANDORA as often done in validation papers – while e.g., number of comparison points or impact of the Lowess(f) monthly running averages is not mentioned/discussed. How much this exercise results would change with a simple mean or median of the daily comparisons? This would allow putting an uncertainty number on the 1.8 and 1.7 PAN/OMI mentioned in page 11.

Consider adding a section or table with the different PANDORA site description, that would help the reader understanding the general differences among the stations (partially already described in the text, but bot for all sites – coordinated of the sites is also missing). This would be a good reference for future studies using these PANDORA data.

Please clarify how some justifying arguments are obtained (add references or explain not shown results). E.g. :

1) P14, line 278 "The relatively moderate TCNO2 values (0.4 to 0.8 DU) are probably a testament to the effectiveness of catalytic converters mandatory on all US automobiles in such a high traffic area" → add reference!;

2) P17, line 290 "The highest amount of TCNO2 recorded during 2018 was about 5DU on 13 July 2018 from 11:20 and 12:30 EST (a time with very light winds (1 km/hr) and moderate temperature (25°C)" → is the meteorology present at each site or only here? Could you shown some correlations? Or is this just a specificity of that time period?

**Specific comments and Technical corrections**

- Line 6: "14 sites" but only 13 are presented – 7 sites in table 1 and 6 in table 2. Same comment for line 13 "Eight additional sites… "

- Line 9 and 11: why mention sites in Northern Hemisphere and Southern hemisphere if this is never mentioned again in the manuscript? Same comment for line 16 "weekly or monthly average basis": weekly comparisons are never mentioned again.
- Line 18 – 19 and 19-22: see general comment, these 2 arguments are not discussed a lot in the paper.
- line 87 – 89: the explanation on how the comparison is done is mixed between this line and lines 165. Consider adding a paragraph grouping all the comparison selection choices (cloud free pixels? What is done with the row anomaly? Why is a 6 minutes time-selection selected for the PANDORA? What type of filtering is done for PANDORA ? (cf mention of impact of clouds in line 130), …
  To my knowledge, the way the selection is done could have an impact on the results (size of OMI pixels, pixels covering the station or not, averaging the ground-based data (mean or median value?), …), and this is only poorly/not discussed. What is the impact of "the Lowess(f) monthly running averages" choice?
- Line 117: change "." to ":". Same for line 137 giving the link to the data: introduce it in a sentence (e.g., Data can be found here: …). Moreover, a table with coordinates and multiple names of the PANDORA stations would be helpful – "waterflow" overpass is e.g. found in the OMI link, but not on the PANDORA link.
- Lines 142-147: give references and refer to this when discussing daily and monthly evolution of fig 7 and 8.
- Line 172-174: why only give an illustration of O3 comparison for Busan? Also in table A1, there are quite some differences in the percent difference from station to station (from 0 in Baltimore to 5.6 in Mauna Loa). How is the PAN/OMI here? Is the largest difference for in O3 also at the same stations than the largest differences for NO2? Is it in stations where we expect most of the NO2 in the stratosphere (Mauna Loa)? How is the NO2 tropo/strato ratio (seen by the satellite?)
  Comment on table A1! (How to explain O3 differences of 2.5 to 2.8% at stations close to surface level?) if not here, at least in the Appendix.
- Figures 3, 4 and 5: in the monthly averages, there is often peaks not seen in OMI (shortly discussed for some stations (lines 179-180 for Busan), but not for all of them. Regularly, there is also a divergent behavior of the monthly average at the edges of the time-series (e.g., end 2016 for Mauna Loa, in 2017 for NASA HQ, end of 2017 for Waterflow, end of 2017 to early 2018 for Boulder ) or OMI columns at the end of the time-series as high as PANDORA (eg Buenos aires, NASA HQ). Is this real of is this related to the "Lowess(f) monthly running averages"?
- Lines 195-196: "The calibration of the Mauna Loa PANDORA will be reviewed as part of a general data quality assurance program that is starting with the most recently deployed PANDORA instruments " - do you mean that the PANDORA data might be off?
- Lines 209-211: there is some repetition with previous paragraphs.
- Tables 1 and 2: add coordinates of the stations and measurement time-periods. How is the "average" among the stations performed? Mean? Median? Does it have a large effect? Consider giving the correlations. Comment on Seoul PAN = 1.2 (more than double of all the other sites!) New York value is missing.
- Line 220: give references of the Discover-AQ campaigns and discuss some of the outcomes (several PADORA on close locations; airborne flights; …) Refer also to other studies dealing

with PANDORA data for validation of NO2, eg., Judd et al., 2019 ( https://www.atmos-meas-tech-discuss.net/amt-2019-161/ ) discussing heterogeneous NO2 situations.

- Lines 245-267: consider re-organizing the paragraphs (order and repetition). Discuss first Fig 6 completely, and then comment on Fig 7. In the comments of figure 6, reference to literature trends is missing (e.g., Duncan et al., 2016; … ). It is a pity that only 6 of the 7 long-term stations are shown in Fig6. Move the discussion of the Boulder trend from the figure caption to the main text. Is there an explanation for the 3 classes of mean bias results ( 1) about -24 to -27% for Boulder, Mauna Loa and NASA HQ; 2) about -37% for Waterflow and 3) about -46% for Buenos Aires and Seoul ) ?
- Lines 258-259: consider giving all the correlation coefficients in the tables as suggested.
- Figure 7 and 8: pity that the figures are not presented for the same year (2018), so that we could compare NASA HQ Washington and New York NO2 levels. Moreover, the TCNO2 axis limit is changing from panel to panel, so it is not so easy to see the seasonal behavior.
- Lines 278-279: "The relatively moderate TCNO2 values (0.4 to 0.8 DU) are probably a testament to the effectiveness of catalytic converters mandatory on all US automobiles in such a high traffic area". Is it purely speculative? Is there any correlation with when the regulation measures have been put in place? Give references!
- Line 284: "the pollution levels are quite high, rivaling the pollution levels in Seoul, South Korea." → this is not seen in Tables 1 and 2, and we don't have these kind of plots for Seoul, only Busan (fig 1).
- Line 293: "For both Washington DC (Fig. 7) and New York City (Fig. 8) there is strong day-to-day and month to month variability that depends on the local weather and the amount of automobile traffic in the area" – has the dependence on weather and traffic been tested or is this a guess or literature reference?
- Line 296: "Poor air quality affecting respiratory health would be improperly characterized by both the OMI average values being too low (Fig. 4) and by missing the extreme pollution events that occur frequently in the late afternoon". Also add a comment (with references) that here total columns are being analyzed, while tropospheric columns could be used, which anyway don't reflect systematically the surface concentrations important for air quality.
- Caption of figure 9: "Lowless(0.08)" it is the first time that the "f" is mentioned. Why is it different than in Herman et al., 2018 (e.g., caption of figure 9 "Lowless(0.1)" )?
- Line 308-309: "there is a period in March 2018 when OMI TCNO2 slightly exceeded that measured by PANDORA." Where are those pixels? Over the sea? What is their size? What is the wind condition?
- Line 2018-2019: "The OMI underestimate is much larger than error estimates for TCNO2 retrievals for either PANDORA or OMI". Consider adding the error on some of the graphs for illustration!
- Add some discussion in the conclusion about new and upcoming satellites (eg TROPOMI with smaller pixels and geostationary that will be able to see the diurnal variation) and the uncertainties of this study (impact of the NASA product selection for OMI (wrt to DOMINO and QA4ECV) and related to the way the comparison is done (see general comment)).
- Appendix: comment on table A1 O3 results (up to 2.8% also outside mountain conditions)
- Referecences: Boersma et al., 2011 is missing. Add suggested references. Mind the formatting!

**Suggested references**:

Duncan, B. N., L. N. Lamsal, A. M. Thompson, Y. Yoshida, Z. Lu, D. G. Streets, M. M. Hurwitz, and K. E. Pickering (2016), A space-based, high-resolution view of notable changes in urban NOx pollution around the world (2005–2014), J. Geophys. Res. Atmos., 121, doi:10.1002/2015JD024121.

Judd, L. M., Al-Saadi, J. A., Janz, S. J., Kowalewski, M. G., Pierce, R. B., Szykman, J. J., Valin, L. C., Swap, R., Cede, A., Mueller, M., Tiefengraber, M., Abuhassan, N., and Williams, D.: Evaluating the impact of spatial resolution on tropospheric $NO_2$ column comparisons within urban areas using high-resolution airborne data, Atmos. Meas. Tech. Discuss., https://doi.org/10.5194/amt-2019-161, in review, 2019.

Boersma, K. F., Eskes, H. J., Richter, A., De Smedt, I., Lorente, A., Beirle, S., van Geffen, J. H. G. M., Zara, M., Peters, E., Van Roozendael, M., Wagner, T., Maasakkers, J. D., van der A, R. J., Nightingale, J., De Rudder, A., Irie, H., Pinardi, G., Lambert, J.-C., and Compernolle, S. C.: Improving algorithms and uncertainty estimates for satellite $NO_2$ retrievals: results from the quality assurance for the essential climate variables (QA4ECV) project, Atmos. Meas. Tech., 11, 6651-6678, https://doi.org/10.5194/amt-11-6651-2018, 2018.

---

## Author Comment (AC3) · 20 Jun 2019

General comments ThemanuscriptpresentsanevaluationoftheOMINO2columnsagainstground-based observations at different sites using Pandora measurements. The authors find that OMI underestimates as expected the GB measurements and they attribute this underestimation to retrieval issues and differences in field of view. They also discuss the effect of NO2 daily cycle. The results are a good addition to the existing literature but their presentation and the way they reach the conclusions might be improved quite a bit as I suggest below.

Specific comments

1. Abstract: L8-15 Should this description of the sites be here in the abstract? Maybe you could write in a more concise way and focus on the results here instead...

**Revised with conclusions moved to the front**

2. L33 Perhaps a reference here, e.g. Krotkov et al. (2016) Krotkov, N. A., McLinden, C. A., Li, C., Lamsal, L. N., Celarier, E. A., Marchenko, S. V., Swartz, W. H., Bucsela, E. J., Joiner, J., Duncan, B. N., Boersma, K. F., Veefkind, J. P., Levelt, P. F., Fioletov, V. E., Dickerson, R. R., He, H., Lu, Z., and Streets, D. G.: Aura OMI observations of regional SO2 and NO2 pollution changes from 2005 to 2015, Atmos. Chem. Phys., 16, 4605-4629, https://doi.org/10.5194/acp-16-4605-2016, 2016.

**Some of these references are now included**

3. L36 you mean "OMI TCNO2 underestimation"?
==The underestimate of OMI $TCNO_2$ at the overpass time compared to ground-based measurements has been previously reported ….==

4. L42 Maybe you can rewrite this more specifically e.g. mentioning that OMI does not capture higher values occurring after the overpass time and thus cannot be used alone for estimating the hazard related to bad AQ.

5. L116-117 Are these overpass files based on the minimum distance between pixel center and the GB site? There is also the possibility to use the pixel actually including the GB site; this might be not the same than the one with the minimum distance from the pixel center. Did you check that? You might also want to analyse the large pixels separately (the ones on the sides of the swath are significantly larger than in nadir) and see if the underestimation is actually related to the size of the pixel and how.

**I have analyzed the data by restricting the distance to less than 30 km (I present one example of such an analysis, Fig. 9) and obtained almost the same results. Most analysis is done in comparison to models using gridded mapped data. Such analysis totally ignores OMI pixel size. There simply is not enough nadir data to analyze.**

6. Fig. 3 first panel: Because of the long time period this plot is really busy and doesn't add much to the one with the monthly data on the right side: maybe you could think to replace it with a scatterplot instead? Same for fig. 4 and 5.

**I disagree with the reviewer. Scatter plots usually hide the key results since they are not time ordered. The best results from a scatter plot are an estimate of the correlation coefficient. I have added the $r^2$ values in Table 1 showing poor correlation as expected. Graphs showing the raw data alongside of monthly averages let the reader see what has been measured.**

7, L245-249 and L261-267 There seems to be a repetition here

**Fixed see page 11 last paragraph**

8. Fig. 6 Could a similar picture be done for the rel. dif. as a function of the OMI pixels size? This might help supporting your conclusion that the underestimations due to the large FOV of OMI. (see also point 5)

**My conclusion about pixel size is confirmed by Judd et al. (2019) quoted in the paper twice (see page 2 and page 19)**

9. Fig. 6 and 9 Can you explain why do you expect from these trend plots? Why do you think the rel. difference should change?

**Without modelling work, I think that for Boulder the suburbs have grown over the past 14 years increasing the amount seen by OMI's larger FOV. This is probably true for NASA HQ and Seoul.**

10. Fig. 7-8 These 3D plots are maybe not so clear if you want to compare the daily cycles in different months: maybe you could replace them with a pcoloror contour-type of plots or even better adding a 1D plot with the mean daily cycles for each month. You could be able to better visualize seasonal differences in the daily cycle. Again, about the daily cycle, you could compare your results with this paper by Boersma et al. 2009, where the seasonal changes in the NO2 daily cycle were analyzed in details. https://www.atmos-chem-phys.net/9/3867/2009/acp-9-3867-2009.pdf

**The 3-D plots were a useful way to present a full year of daily data for a given site and simply indicate the time of the OMI observation relative to the high NO2 values. I tried color contour plots. Those work also, but are less dramatic and have no extra information compared to the 3-D plot. The peaks are very obvious in the 3-D plot without referring to a color scale.**

11. Summary: You could add a couple of sentences on the potential of the new retrievals from TROPOMI (much smaller pixel) as well as TEMPO higher (hourly) temporal resolution.

**TEMPO and TropOMI are now mentioned for time resolution and reduced pixel size**

Technical comments L30 foe -> for   **Good catch**

L169 PANDRA -> PANDORA   **Thank you**

L209-211 This is a bit of a repetition. **You are right, but it is a small repetition**

[revised manuscript text omitted]

---

## Author Comment (AC4) · 20 Jun 2019

PLEASE SEE A REVISED VERSION OF THE PAPER AT THE END OF THESE COMMENTS THAT INCORPORATES THE CHANGES. YELLOW=REV 1 GREEN=REV2 GREY=AUTHOR CHANGES

**Reviewer #2**

This paper presents a comparison between OMI NASA v3.1 and PANDORA total NO2 VCD, showing a clear under-estimation of the OMI data at 7 long-term sites and 6 campaign-based sites. The results at most of the sites are presented and discussed and few arguments for the general underestimation result are mentioned. Although the paper is interesting and fulfill the scope of AMT, there is a lack of reference to literature (previous similar studies and scientific proof/reference of why such differences at the different sites). Sensitivity tests or further comparisons on OMI pixel sizes (edge and center of the swath, different position of the pixels, GB time-selection) could be done to help justifying the proposed conclusion. I recommend the publication after the suggested revisions.

General comments:

The paper is short and easy to read, but it lack some "proof" of the proposed explanation of the OMI under-estimation (argument

1= "Because of the local inhomogeneity of NO2 emissions, the large OMI FOV is the most likely factor when comparing OMI TCNO2 to retrievals from the small PANDORA effective FOV", line 20 and argument.

See page 2 Judd et al., 2018; Nowlan et al., 2016

2= "OMI estimated air mass factor, surface reflectivity, and the OMI 24x13 km2 FOV (field of view) are three factors that can cause OMI to underestimate TCNO2", line 18).

See the references Boersma et al., 2011; Lin et al., 2015; Nowlan et al., 2016; Lorente et al., 2018

**Krotkov et al., 2017**

Some sensitivity tests on the how much the choices made for the OMI pixel selection (FOV distance d<5km for an SZA<70, line 165) and PANDORA selection ("daily data matched to the OMI overpass times ±6miutes", line 87) affect the results would support argument.

**It is the Pandora FOV is less than 5 km from the Pandora site for SZA <70 degrees. This is simple geometry. I have added a comment for this.**

1 (or at least give an uncertainty range). Additional comparison (or at least further comments on other OMI retrievals, such as DOMINO (Boersma et al., 2011) or QA4ECV (Boersma et al., 2018)), would support argument

The DOMINO algorithm has some known problems (see reference) and the QA4ECV results are very similar to the NASA results. Because of this, I have put in a statement about the QA4ECV results and a reference.

2. Moreover, a lot of (redundant) figures are given (daily and monthly panels in Fig 3, 4, 5 and 9) could be simplified by plotting the mean and the variability – or a scatter plot of OMI vs PANDORA as often done in validation papers – while e.g., number of comparison points or impact of the Lowess(f) monthly running averages is not mentioned/discussed. How much this exercise results would change with a simple mean or median of the daily comparisons? This would allow putting an uncertainty number on the 1.8 and 1.7 PAN/OMI mentioned in page 11.

**I disagree with referee about the redundancy. Figure 3A shows the daily data and Figure 3B shows the averages. Even though both show the difference, it is useful to see the daily data.**

Consider adding a section or table with the different PANDORA site description, that would help the reader understanding the general differences among the stations (partially already described in the text, but bot for all sites – coordinated of the sites is also missing). This would be a good reference for future studies using these PANDORA data.

**OK See Table 2**

Please clarify how some justifying arguments are obtained (add references or explain not shown results). E.g. :

P14, line 278 "The relatively moderate TCNO2 values (0.4 to 0.8 DU) are probably a testament to the effectiveness of catalytic converters mandatory on all US automobiles in such a high traffic area"

**Gary A. Bishop and Donald H. Stedman, Reactive Nitrogen Species Emission Trends in Three Light-/Medium-Duty United States Fleets, *Environmental Science & Technology* 2015 *49* (18), 11234-11240, DOI: 10.1021/acs.est.5b02392**

add reference!;

 P17, line 290 "The highest amount of TCNO2 recorded during 2018 was about 5DU on 13 July 2018 from 11:20 and 12:30 EST (a time with very light winds (1 km/hr) and moderate temperature (25°C)"

☑ is the meteorology present at each site or only here? Could you shown some correlations? Or is this just a specificity of that time period?

Meteorology affects the amount of NO2 observed at all sites. I described the meteorology for this site on a 13 July 2018 because the amount, 5DU, was very unusual. In general, days with no winds show high values of NO2 near the sources for NO2.

Harkey, M., Holloway, T., Oberman, J., and Scotty, E., An evaluation of CMAQ NO2 using observed chemistry-meteorology correlations, J. Geophys. Res. Atmos., 120, 11,775–11,797, doi:10.1002/2015JD023316, 2015

Specific comments and Technical corrections

- Line 6: "14 sites" but only 13 are presented – 7 sites in table 1 and 6 in table 2. Same comment for line 13 "Eight additional sites..."

**Corrected in Table 2**

- Line 9 and 11: why mention sites in Northern Hemisphere and Southern hemisphere if this is never mentioned again in the manuscript? Same comment for line 16 "weekly or monthly average basis": weekly comparisons are never mentioned again. -

**Now NH and SH mentioned in the text on Page 6. Even though true, I removed weekly from line 16.**

Line 18 – 19 and 19-22: see general comment, these 2 arguments are not discussed a lot in the paper. –

Surface reflectivity was discussed on page 2. I added: "Accurately determining the AMF for TCNO2 requires a-priori knowledge of the NO2 profile shape, which is estimated from coarse resolution model calculations (Boersma et al., 2011)," The references give extensive discussions of these factors and their effects.

line 87 – 89: the explanation on how the comparison is done is mixed between this line and lines 165. Consider adding a paragraph grouping all the comparison selection choices (cloud free pixels? What is done with the row anomaly? Why is a 6 minutes time-selection selected for the PANDORA? What type of filtering is done for PANDORA ? (cf mention of impact of clouds in line 130), ...

The following paragraph has been added to page 3

OMI overpass data, https://avdc.gsfc.nasa.gov/index.php?site=666843934&id=13, are filtered for the row anomaly and cloudy pixels. The selection of a  $\pm 6$  minute window represents 720 seconds or 9 PANDORA measurements averaged together around the OMI overpass time to reduce the effect of any outlier points. PANDORA makes an NO2 measurement every 80 seconds. The specific value of  $\pm 6$  minutes is arbitrary but increases the effective signal to noise ratio by a factor of 3. PANDORA data are filtered for significant cloud cover by examining the effective variance in sub-interval (20 seconds) measurements. Each PANDORA listed measurement is the average of up to 4000 (clear sky) individual measurement made over 20 seconds.

To my knowledge, the way the selection is done could have an impact on the results (size of OMI pixels, pixels covering the station or not, averaging the ground-based data (mean or median value?), ...), and this is only poorly/not discussed. What is the impact of "the Lowess(f) monthly running averages" choice? -

The overpass data set represents the closest filtered OMI pixel to the specified site on a given day. On any single OMI measurement, the OMI FOV may not be exactly over the site. This is an intrinsic characteristic of OMI data as used for practical purposes. An alternative would be to use gridded data on a fixed latitude x longitude grid. The result is an even wider area view of the specified site (an average of more OMI pixels). The point is that OMI data are used to represent the amount of NO2 over a given location whether in comparison to PANDORA or a model study. Air quality decisions are made based on OMI data for urban and unpolluted regions that include intrinsic area averaging.

The impact of Lowess or adjacent averaging over a month's worth of data is to smooth out the daily variation and show an average difference. Daily data are presented as well. Weekly averages would show the same qualitative result. Lowess is preferable to adjacent averaging, since it is least-squares weighted and reduces the effect of possible outlier points

Line 117: change "." to ":". **Fixed** Same for line 137 giving the link to the data: introduce it in a sentence (e.g., Data can be found here: ...**It already says that**). Moreover, a table with coordinates and multiple names of the PANDORA stations would be helpful – "waterflow" overpass is e.g. found in the OMI link, but not on the PANDORA link. **Waterflow is labelled Four Corners on the website – I have added this name to the paper.**

Lines 142-147: give references and refer to this when discussing daily and monthly evolution of fig 7 and 8. –

(Lamsal et al., 2013; Bechle et al., 2013).

Line 172-174: why only give an illustration of O3 comparison for Busan?

I could give O3 plots for all sites at the expense of more figures. However, the appearance is very similar to that for Busan (except Mauna Loa because of altitude effects). The results for all sites are summarized in Table A1. The purpose is to show that all instruments were working properly.

Also in table A1, there are quite some differences in the percent difference from station to station (from 0 in Baltimore to 5.6 in Mauna Loa).

The Mauna Loa difference is caused by altitude for O3 with Pandora missing the lowest 3.4 km. NO2 differences are not related to ozone differences. This is stated in Table A1. The differences are not a function of the PANDORA instruments nor the retrieval algorithms.

For O3, the biggest error is the lack of effective O3 temperature in the algorithm. An average effective O3 temperature is used instead of a measured temperature. An example of this is give in Herman et al., 2015; 2017 for Boulder Colorado

How is the PAN/OMI here? Is the largest difference for in O3 also at the same stations than the largest differences for NO2? **No**

Is it in stations where we expect most of the NO2 in the stratosphere (Mauna Loa)? How is the NO2 tropo/strato ratio (seen by the satellite?)

For Mauna Loa, the Pandora saw more NO2 than is possible in the stratosphere. The NO2 is drifting upward from the coastal areas. This is mentioned in the paper.

Comment on table A1! (How to explain O3 differences of 2.5 to 2.8% at stations close to surface level?) if not here, at least in the Appendix. –

Without proof, I suspect that the incorrect average effective temperature is the cause of a part of the difference as it was at Boulder Colorado, since we use temperature dependent ozone cross sections for both Pandora and OMI. There is also the issue of field calibration to remove the reference amount of ozone (modified Langley calibration) for zero airmass. This is discussed in an earlier paper and not part of the scope of this paper. This procedure has not been done for City College nor for HUFS. If the instruemnts were not operating properly (e.g., pointing at the sun), the differences would be much larger.

Figures 3, 4 and 5: in the monthly averages, there is often peaks not seen in OMI (shortly discussed for some stations (lines 179-180 for Busan), but not for all of them. Regularly, there is also a divergent behavior of the monthly average at the edges of the time-series (e.g., end 2016 for Mauna Loa, in 2017 for NASA HQ, end of 2017 for Waterflow, end of 2017 to early 2018 for Boulder ) or OMI columns at the end of the time-series as high as PANDORA (eg Buenos aires, NASA HQ). Is this real of is this related to the "Lowess(f) monthly running averages"? – '

**I should exclude endpoints for running averages. I will change the figures. (NOT DONE YET)**

Lines 195-196: "The calibration of the Mauna Loa PANDORA will be reviewed as part of a general data quality assurance program that is starting with the most recently deployed PANDORA instruments " - do you mean that the PANDORA data might be off? -

No, there is a new Pandora installed at Mauna Loa after the sun tracker broke down. At the time of this writing, data from the new Pandora are not used. The sentence has been changed.

Recently, the original Mauna Loa PANDORA has been replaced. The new instrument's calibration will be reviewed as part of a general data quality assurance program that is starting with the most recently deployed or upgraded PANDORA instruments at about 100 locations.

Lines 209-211: there is some repetition with previous paragraphs. -

**This paragraph has been moved (Page 9)**

Tables 1 and 2: add coordinates of the stations and measurement time-periods. How is the "average" among the stations performed? Mean? Median? Does it have a large effect? Consider giving the correlations. Comment on Seoul PAN = 1.2 (more than double of all the other sites!) New York value is missing. –

The average values are simply the arithmetic average of the daily points for each location. The overall average is the arithmetic average of the above averages. Seoul is the most polluted city considered, so

the average value is higher. However, the ratio with OMI is similar to most of the other sites. New York has been added to Table 2.

I have added correlation coefficients to Table 1 and the sentence on page 9

For example, the PANDORA at NASA Headquarters in Washington DC tracks the OMI measurement quite well on a monthly average basis with a correlation coefficient of  $r^2(mn) = 0.7$  even though the daily correlation is low ( $r^2(dy) = 0.17$ ). Other sites have only short periods of correlation and overall weak correlation (Table 1 showing daily, dy and monthly, mn, correlation coefficients for the graphs in Figures 4 and 5)

Line 220: give references of the Discover-AQ campaigns and discuss some of the outcomes (several PADORA on close locations; airborne flights; ...) Refer also to other studies dealing with PANDORA data for validation of NO2, eg., Judd et al., 2019 ( https://www.atmos-meastech-discuss.net/amt-2019-161/ ) discussing heterogeneous NO2 situations. –

**The Judd at al reference has been added on Page 2 and backs up the thesis that spatial resolution is a major cause of the underestimate by OMI compared to PANDORA.**

Judd, L. M., Al-Saadi, J. A., Janz, S. J., Kowalewski, M. G., Pierce, R. B., Szykman, J. J., Valin, L. C., Swap, R., Cede, A., Mueller, M., Tiefengraber, M., Abuhassan, N., and Williams, D.: Evaluating the impact of spatial resolution on tropospheric NO2 column comparisons within urban areas using high-resolution airborne data, Atmos. Meas. Tech. Discuss., https://doi.org/10.5194/amt-2019-161, in review, 2019.

Lines 245-267: consider re-organizing the paragraphs (order and repetition). Discuss first Fig 6 completely, and then comment on Fig 7. In the comments of figure 6, reference to literature trends is missing (e.g., Duncan et al., 2016; ... ).

Duncan et al. (2016) estimated trends from OMI TCNO2 time series and found that the Seoul metropolitan area had a decrease of  $-1.5 \pm 1.3$  %/Year (2005 – 2014) consistent with OMI estimated change of  $-1.4 \pm 1$ %/year (2012 -2018) in this paper. However, for the small area near Yonsei University, the decrease estimated from PANDORA is  $-5.8 \pm 0.75$  %/Year. Park (2019) estimates that metropolitan Seoul has decreased in population even as surrounding areas have increased population.

**(see page 12)**

It is a pity that only 6 of the 7 long term stations are shown in Fig 6.

While not showing an extra plot in Fig. 6 I have added the results for the 7th long-term site, Busan, in the text,

The results for Busan (from Fig. 3) show a least squares average for the percent difference of -48  $\pm$  0.8% for the 2012 – 2018 period with a slope of 6.8  $\pm$  1%/Year. There is a decrease in the percent difference after October 2015 (Fig. 3) that is mainly from PANDORA seeing less TCNO2

than during the 2012 – 2014 period. There is a gap in the Busan time series from July 2014 until April 2015 when the original PANDORA was replaced with a new instrument. The calibrations of both PANDORAS appear to be correct. Because of the break in the time series it is not clear whether there was a change in local conditions around Pusan University compared to the wide area observed by OMI.

Move the discussion of the Boulder trend from the figure caption to the main text.

**Done**

Is there an explanation for the 3 classes of mean bias results (1) about -24 to -27% for Boulder, Mauna Loa and NASA HQ; 2) about -37% for Waterflow and 3) about -46% for Buenos Aires and Seoul )? –

I do not know the explanation for the differences between the narrow view trends (PANDORA and the wide area trends (OMI). The other long-term site considered, Busan, has gaps in the data record that are fairly large.

I added (page 14)

For some sites (see Fig. 6), PANDORA and OMI trends are the same (Waterflow, NM, Buenos Aires, and Mauna Loa) while the other 3 sites show significantly different trends (Boulder, NASA HQ, and Seoul).

Lines 258-259: consider giving all the correlation coefficients in the tables as suggested.

**see Table 1**

- Figure 7 and 8: pity that the figures are not presented for the same year (2018), so that we could compare NASA HQ Washington and New York NO2 levels.

**I do not have a complete data record for NASA HQ in 2018 and only have 2018 for New York City**

Moreover, the TCNO2 axis limit is changing from panel to panel, so it is not so easy to see the seasonal behavior. –

Making all of the scales the same will obscure the behavior relative to the OMI overpass time, which is the subject of this paper.

There is no easy way to represent the seasonal behavior vs time of day on a minute by minute basis or even an hourly basis for such complex highly variable behavior of TCNO2 shown in Figures 7 and 8. The seasonal variation at the OMI overpass time is given in Fig. 9 for NYC and in Fig. 4 for NASA HQ. The general seasonal variation is not the subject of this paper. However, while not part of this paper, I have added a graph at the end of this reply that shows the monthly average behavior of TCNO2 for CCNY for four different times of the day 10:00, 13:50, 14:00, 16:00. With variations in magnitude, the seasonal behavior is similar for the different times of the day.

Lines 278-279: "The relatively moderate TCNO2 values (0.4 to 0.8 DU) are probably a testament to the effectiveness of catalytic converters mandatory on all US automobiles in such a high traffic area". Is it purely speculative? Is there any correlation with when the regulation measures have been put in place? Give references! – **(Bishop and Steadman, 2015).**

Line 284: "the pollution levels are quite high, rivaling the pollution levels in Seoul, South Korea." It is not seen in Tables 1 and 2, and we don't have these kind of plots for Seoul, only Busan (fig 1). –

**I added (see Fig. 5)**

Line 293: "For both Washington DC (Fig. 7) and New York City (Fig. 8) there is strong day-today and month to month variability that depends on the local weather and the amount of automobile traffic in the area" – has the dependence on weather and traffic been tested or is this a guess or literature reference? –

Seo, J., Park, D.-S. R., Kim, J. Y., Youn, D., Lim, Y. B., and Kim, Y.: Effects of meteorology and emissions on urban air quality: a quantitative statistical approach to long-term records (1999–2016) in Seoul, South Korea, Atmos. Chem. Phys., 18, 16121-16137, https://doi.org/10.5194/acp-18-16121-2018, 2018.

Zheng, G. J., Duan, F. K., Su, H., Ma, Y. L., Cheng, Y., Zheng, B., Zhang, Q., Huang, T., Kimoto, T., Chang, D., Pöschl, U., Cheng, Y. F., and He, K. B.: Exploring the severe winter haze in Beijing: the impact of synoptic weather, regional transport and heterogeneous reactions, Atmos. Chem. Phys., 15, 2969–2983, https://doi.org/10.5194/acp-15-2969-2015, 2015.

Md. Shohel Reza Amin, Umma Tamima, and Luis Amador Jimenez, "Understanding Air Pollution from Induced Traffic during and after the Construction of a New Highway: Case Study of Highway 25 in Montreal," Journal of Advanced Transportation, vol. 2017, Article ID 5161308, 14 pages, https://doi.org/10.1155/2017/5161308, 2017

Andersen, M. Hvidberg, S.S. Jensen, M. Ketzel, S. Loft, M. Sørensen, A. Tjønneland, K. Overvad, O. Raa schou-Nielsen Chronic obstructive pulmonary disease and long-term exposure to traffic-related air pollution: A cohort study, Am. J. Respir. Crit. Care Med., 183, pp. 455-461, 10.1164/rccm.201006-09370C, 2011.

Line 296: "Poor air quality affecting respiratory health would be improperly characterized by both the OMI average values being too low (Fig. 4) and by missing the extreme pollution events that occur frequently in the late afternoon". Also add a comment (with references) that here total columns are being analyzed, while tropospheric columns could be used, which anyway don't reflect systematically the surface concentrations important for air quality. –

Page 20 It should be noted that TCNO2 does not accurately represent the NO2 concentration at the surface, since it is mostly a measure of the amount in the lower 2 km. However, it is roughly proportional to the surface measurements close to the pollution sources (Bechle et al., 2013; Knepp et al., 2014) with the proportionality dependent on the profile shape near the ground.

Caption of figure 9: "Lowless(0.08)" it is the first time that the "f" is mentioned. Why is it different than in Herman et al., 2018 (e.g., caption of figure 9 "Lowless(0.1)")? –

I could have given the f-value for each graph. It is the fraction f of data points over which the Lowess(f) algorithm is applied to form an average local least squares fit. This is similar to the number of points included in an arithmetic running average. The exact fraction will depend on the number of points in a month's worth of data compared to the entire data set.

Line 308-309: "there is a period in March 2018 when OMI TCNO2 slightly exceeded that measured by PANDORA." Where are those pixels? Over the sea? What is their size? What is the wind condition? -

The OMI pixels for the March 2018 period are distributed over both land and water. I have replotted the data only using points less than 30 km from CCNY. The results are very similar, but not identical to when D

Figure 9 has been replaced to exclude pixels further than 30 km. The results are almost identical. Most papers comparing OMI data with models related to air quality estimates use a gridded version of OMI data totally ignoring OMI pixel size in order to produce local area maps of TCNO2

Line 2018-2019: "The OMI underestimate is much larger than error estimates for TCNO2 retrievals for either PANDORA or OMI". Consider adding the error on some of the graphs for illustration!

**I added the error estimates for the least squares mean percent differences to the graphs in Fig. 6.**

-

Add some discussion in the conclusion about new and upcoming satellites (eg TROPOMI with smaller pixels and geostationary that will be able to see the diurnal variation)

**Done**

and the uncertainties of this study (impact of the NASA product selection for OMI (wrt to DOMINO and QA4ECV) and related to the way the comparison is done (see general comment)).

**See page 2**

- Appendix: comment on table A1 O3 results (up to 2.8% also outside mountain conditions) -

The 2.8% offset is too large since the PANDORA calibration looks very good. Both data sets track each other quite well with high correlation on a monthly average basis. The most likely cause is an improper effective ozone temperature correction for PANDORA that was obtained from a model calculation

References: Boersma et al., 2011 is missing. Add suggested references. Mind the formatting!

**Added**

Suggested references:

Duncan, B. N., L. N. Lamsal, A. M. Thompson, Y. Yoshida, Z. Lu, D. G. Streets, M. M. Hurwitz, and K. E. Pickering (2016), A space-based, high-resolution view of notable changes in urban NOx pollution around the world (2005–2014), J. Geophys. Res. Atmos., 121, doi:10.1002/2015JD024121.

Judd, L. M., Al-Saadi, J. A., Janz, S. J., Kowalewski, M. G., Pierce, R. B., Szykman, J. J., Valin, L. C., Swap, R., Cede, A., Mueller, M., Tiefengraber, M., Abuhassan, N., and Williams, D.: Evaluating the impact of spatial resolution on tropospheric NO2 column comparisons within urban areas using high-resolution airborne data, Atmos. Meas. Tech. Discuss., https://doi.org/10.5194/amt-2019-161, in review, 2019.

Boersma, K. F., Eskes, H. J., Richter, A., De Smedt, I., Lorente, A., Beirle, S., van Geffen, J. H. G. M., Zara, M., Peters, E., Van Roozendael, M., Wagner, T., Maasakkers, J. D., van der A, R. J., Nightingale, J., De Rudder, A., Irie, H., Pinardi, G., Lambert, J.-C., and Compernolle, S. C.: Improving algorithms and uncertainty estimates for satellite NO2 retrievals: results from the quality assurance for the essential climate variables (QA4ECV) project, Atmos. Meas. Tech., 11, 6651-6678, https://doi.org/10.5194/amt-11-6651-2018, 2018.

---

## Author Response (AR2)

**Associate Editor Decision: Publish subject to minor revisions (review by editor) (16 Aug 2019) by Folkert Boersma Comments to the Author: Dear authors,**

After carefully reading the review reports and your response and revision to the manuscript, I think the paper can be accepted for publication in AMT, but I would like you to make the following modifications. Congratulations on a nice and comprehensive piece of work.

In the abstract of the revised manuscript (L11-12) you write "OMI always misses the frequently much higher values of TCNO2 that occur after the OMI overpass time". This should be rephrased as OMI is designed such that -by definition- it cannot measure TCNO2 after its overpass.

**The new abstract appears at the end of this reply**

The next sentence should be nuanced or clarified. You write "OMI retrieved TCNO2 are not suitable for air quality assessments as related to human health, especially in polluted urban areas", but this statement is too sweeping. Please write exactly what you can justify based on your study, which is that OMI cannot resolve NO2 pollution within a city, which makes it less suitable for urban surface NO2 assessments (it can still provide some useful information if there is nothing else!).

Compared to local ground-based or aircraft measurements, OMI cannot resolve spatially variable TCNO2 pollution within a city or urban areas, which makes it less suitable for air quality assessments related to human health.

**L22: NO2 emissions --> NOx emissions Changed**

L43-44: "First, the mid-day OMI observations do not see the large diurnal variation of TCNO2 that usually occur after the 13:30 overpass time". Please rephrase such to make clear that OMI cannot be expected to see diurnal variation in the first place (unless when combined with another instrument, or at high latitudes).

**Because of OMI's selected polar orbit, it is not possible for the mid-day OMI observations to see the large diurnal variation of TCNO2 that usually occur after the 13:30 overpass time,**

L177: "In addition to missing the TCNO2 diurnal variation": again this should be reformulated. Readers should not get the impression that OMI was supposed to capture the TCNO2 diurnal variation.

**In addition to not being designed to observe the TCNO2 diurnal variation**

L219-220: "The PANDORA values suggest upward airflow from the nearby circumferential ring road and resort areas". This statement comes a bit out of the blue. How do PANDORA values suggest this, why is it important? Please clarify.

Reply: There are no emission or combustion sources of NO2 at MLO at 3.4 km altitude. However, there is considerable production of NO2 arising from power plants and automobile traffic near sea level. This implies that somehow NO2 is getting to 3.4 km. Since the Pandora values are in excess of stratospheric values<mark>,</mark> to me this suggests upward airflow carrying NO2 from its sources. <mark>See the figure on pare 2 of this reply</mark>

The sentence has been changed to read:

OMI, which mainly measures values over the clean ocean, has an average value of about 0.1 DU (see appendix Fig. A2). Since there are no emission or combustion sources of NO2 at high altitudes near MLO at 3.4 km, the PANDORA values suggest upward airflow from the near sea level circumferential ring road, Keahole oil Power plant, and resort areas.

Regarding the comments of Rev#2 and my earlier comments: I still think that Fig. 7 and 8 should be improved. They are not very clear, and require a lot of effort from a reader to see where the OMI time sits in the 3-D landscape. As suggested earlier, there are better ways to show the OMI 'underestimate' in the diurnal pattern. If you feel you really want to stick to Figure 7 and 8, please do so, but I think it hurts the clarity of the manuscript.

Last but not least, the thing still "hanging" with this work is the diurnal cycle in NO2 observed by the PANDORA, which often shows strong increases or maxima in the afternoon. Based on knowledge of the diurnal cycle in emissions, chemistry, and BL development this is quite surprising, and does not sit well with me. It would mean that we are totally overlooking an important process in the air pollution models, or that emissions are very, very different from what we think we know.

I suspect the sources are not well specified in urban areas. When I worked with the EPA models, the sources were very crude approximations to a very complex emission system.

But of course it could also mean that the PANDORA measurements have a weakness or not yet understood sensitivity. In the afternoon, the Sun starts to set, aerosols have been building up, and radiative transfer may not be as easy as in the morning or early afternoon, and this might affect the PANDORA retrieved values in the late afternoon.

Aerosols without spectral structure have very little effect on direct-sun measurements and a DOAS style retrieval. The main effect of heavy aerosol loading is to decrease the signal to noise ratio for PANDORA. In a previous paper, I showed that even the presence of moderate cloud cover does little to degrade the retrieval.

From my AMT publication the retrieval with cloud cover is barely disturbed down to a reduction in signal of about 8. Herman et al., Atmos. Meas. Tech., 11, 4583–4603, 2018 https://doi.org/10.5194/amt-11-4583-2018

Added on page 8

Figure 6 also illustrates TCNO2 diurnal behavior at two other sites, NASA HQ in Washington, DC and at City College of New York and compares the values to the OMI retrieved TCNO2.

Both Figs. 6 and 2A show the PANDORA TCNO2 retrieval with the values of the SZA plotted on the same graph showing that the direct-sun retrievals are good out to SZA = 70°. Depending on atmospheric conditions, retrievals using BEER's law absorption attenuation and spectral fitting for SZA > 75° begin to yield non-physical values (TCNO2 too small). During mid-day measurements, the signal to noise ratio is very high since over 4000 clear-sky measurements are averaged together to produce one data point every 20 seconds. Even with aerosol loading or moderate cloud cover blocking the sun, the retrievals are still accurate (Herman et al., 2018).

Figure 2. (a)  $C(NO_2)$  amounts from Pandora 27 and 35 in Yeoju, Korea during 3 June 2016 and their difference |Pan35-Pan27| < 0.05 DU. (b) Pandora 35 estimate of cloud or aerosol reduced measured counts s-1 at approximately 500 nm.

The manuscript needs at least a short paragraph discussing strengths/weaknesses/unknowns of the PANDORA method, also in view of other publications addressing the issue of diurnal variation in NO2. One obvious question that comes to mind is whether PANDORA measurements in the late afternoon have been validated or evaluated themselves (against MAX-DOAS). The other one is the concern on whether PANDORA does not measure too high NO2 at Mauna Loa ("more than possible in the stratosphere") - a further indication that something may be amiss with late afternoon measurements.

I have added some text to the paper discussing accuracy and precision and references to two previous papers.

The accuracy and precision of PANDORA TCNO2 measurements has been previously discussed (Herman et al., 2009; 2018).

Page 7 line 244 An example of the diurnal behavior of TCNO2 at Waterflow, New Mexico on 6 June 2012 is shown in Fig. 6 to illustrate the behavior of PANDORA TCNO2 retrievals at a wide range of SZA. The terrain surrounding the Pandora site is flat with no obstructions (buildings) permitting observations to very high SZA. Almost every day the power plant briefly puts out very high emissions of NO2 as part of its daily boiler cleaning cycle. This can be seen in the very high peak value of TCNO2 of 3.4 DU compared to the nominal value of 0.5 DU occurring for most of the day. The value from the OMI retrieval at 21:01 GMT (14:01 local standard time) is about 0.2 DU compared to the PANDORA value of about 0.5DU. Figure 6 also illustrates TCNO2 diurnal behavior at two other sites, NASA HQ in Washington, DC and at City College of New York and compares the values to the OMI retrieved TCNO2.